# SPARSE MOE WITH LANGUAGE-GUIDED ROUTING FOR MULTILINGUAL MACHINE TRANSLATION

**Xinyu Zhao[1]**    **Xuxi Chen[2]**    **Yu Cheng[3]**    **Tianlong Chen[1,4,5]**
[1]The University of North Carolina at Chapel Hill    [2]The University of Texas at Austin
[3]The Chinese University of Hong Kong
[4]MIT    [5]Harvard University
{xinyu,tianlong}@cs.unc.edu
xxchen@utexas.edu   chengyu@cse.cuhk.edu.hk

## ABSTRACT

Sparse Mixture-of-Experts (SMoE) has gained increasing popularity as a promising framework for scaling up multilingual machine translation (MMT) models with negligible extra computational overhead. However, current SMoE solutions neglect the intrinsic structures of the MMT problem: (*a*) *Linguistics Hierarchy*. Languages are naturally grouped according to their linguistic properties such as language families, phonological features, *etc*; (*b*) *Language Complexity*. Learning difficulties vary for different languages due to their available resources, grammar complexity *etc*. Therefore, routing a fixed number of experts (*e.g.*, 1 or 2 experts in usual) only at the word level leads to inferior performance. To fill in the missing puzzle, we propose `Lingual-SMoE` by equipping the SMoE with adaptive and linguistics-guided routing policies. Specifically, it (1) extracts language representations to incorporate linguistic knowledge and uses them to allocate experts into different groups; (2) determines the number of activated experts for each target language in an adaptive and automatic manner, according to their difficulty level determined by data abundance, which aims to mitigate the potential over-/under-fitting problems of learning easy/difficult translations. Sufficient experimental studies on MMT benchmarks with {16, 50, 100} languages and various network architectures, consistently validate the superior performance of our proposals. For instance, `Lingual-SMoE` outperforms its dense counterpart by over 5% BLEU scores on the `OPUS-100` dataset. [1]

## 1 INTRODUCTION

Multilingual Machine Translation (MMT) aims to resolve multiple translation directions simultaneously in one unified model and has attracted considerable attention in both academia and industry. Aharoni et al. (2019); Johnson et al. (2016); Aharoni et al. (2019) reveal that MMT models are capable of adapting to low-resource scenarios, benefiting from the joint optimization of multiple translations. Nevertheless, as the number of languages involved in MMT increases (*e.g.*, > 50), the story starts to change. As illustrated by Conneau et al. (2019); Sachan & Neubig (2018), the language interference emerges and poses an obstacle to achieving satisfactory performance, which is a troublesome consequence of the competing gradients among different languages. Unfortunately, such a problem will be exacer-

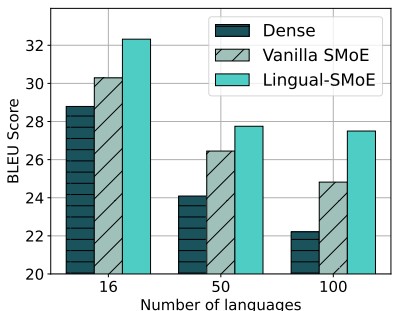

Figure 1: The MMT performance of {dense model, vanilla SMoE, `Lingual-SMoE` (Ours)} on `OPUS-100` with different amount of languages pairs $\in \{16, 50, 100\}$.

bated by severe data imbalance, leading to over-fitting on low-resource translation directions or catastrophic forgetting on previously trained samples (Lakew et al., 2018; Elbayad et al., 2023).

---

[1]Our code is provided at https://github.com/UNITES-Lab/Lingual-SMoE.

A natural remedy is to scale up the model capacity, which has been demonstrated as an effective way to improve multilingual machine translation (Shazeer et al., 2018; Radford & Narasimhan, 2018; Devlin et al., 2019). However, these gigantic language models with massive parameter counts are extremely computationally intensive. For example, training the popular GPT-based model (Brown et al., 2020) with billions of parameters typically requires thousands of GPU days. To provide an efficient alternative, pioneering researchers introduce the Sparsely-gated Mixture-of-Experts (SMoE) framework (Lepikhin et al., 2021; Fedus et al., 2021; Shazeer et al., 2017; Zoph et al., 2022), which is designed with an input-dependent conditional computing fashion. Specifically, SMoE only activates the relevant model pieces given an input sample, and this dynamic sparse computing facilitates the training of huge language models with feasible resource costs.

However, most existing practices adapt the SMoE algorithm in a straightforward way and overlook intrinsic structures in the MMT problem, *i.e.*, *Linguistic Hierarchy* and *Heterogeneous Language Complexity*. ① Even incorporating linguistic knowledge in an exemplary way, it helps. For example, Fan et al. (2021); Zhang et al. (2020) customize several language-specific components in the translation model and enjoy an enhanced performance. Kudugunta et al. (2021) takes one step further by routing input samples to different experts in the SMoE based on translation language representations. Although these initial efforts achieve great results, the exploitation of prior linguistic knowledge is highly insufficient. *Languages have their hierarchy.* Specifically, different languages can be organized in a tree structure according to their language family, grammar, phonological features, *etc*. How to leverage such linguistic priors for SMoE design in MMT, is a challenging yet rewarding question. ② Another limitation of current SMoE approaches is the neglect of the heterogeneous complexity of diverse language translations. Due to the variance of factors such as grammatical complexity and available resources, the translation difficulties can vary significantly (Goyal et al., 2021; Team et al., 2022; Heffernan et al., 2022). However, existing SMoEs use a fixed model size (*e.g.*, 1 or 2 experts) to handle all translation directions. It potentially compromises certain translations since excessive or insufficient model capacity could result in over-fitting or under-fitting issues, respectively. In addition, manual adjustments of the expert capacity for each language are laborious and suboptimal due to the interplay among multiple translation objectives. Then, how to take the heterogeneity of language difficulty into consideration to adaptively determine the appropriate network capacity, is a necessary yet beneficial step towards superior SMoE models.

To answer the aforementioned questions, we propose a novel `Lingual-SMoE` for MMT, by designing language-guided routing policies. It allocates experts in a hierarchical manner and enables the language-specific model capacity, which brings significant performance improvements as demonstrated in Figure 1. The advantages of our effective strategy are multi-fold: (1) injecting rich linguistic knowledge to the expert routing process via encouraging experts to specialize in certain language families; (2) mitigating potential over-/under-fitting on translation directions with different difficulty levels determined by data availability by dynamically adjusting the expert number based on training performance. Our contributions can be summarized as follows:

- ⋆ We propose an innovative SMoE framework for multilingual machine translation, *i.e.*, `Lingual-SMoE`, by considering two unique properties of *linguistic hierarchy* and *heterogeneous language complexity* in MMT problems.

- ⋆ We design a hierarchical routing policy in `Lingual-SMoE` that learns to route input samples with multi-granularity information of {language family, language, and token}.

- ⋆ We introduce a dynamic expert allocation mechanism to adaptively determine adequate expert capacity for each language translation with distinctive difficulty levels. Training dynamics are monitored to enable automatic adjustments.

- ⋆ Extensive experiments with different data resources and number of languages consistently evidence the effectiveness of `Lingual-SMoE`. For example, our language-guided routing proposals outperform its dense baseline by a clear performance margin $5\%$ on `OPUS-100`.

## 2 RELATED WORKS

**Multilingual Machine Translation (MMT).** Multilingual machine translation extends neural machine translation to the scenario with multiple language pairs, which is a popular paradigm in natural language processing (NLP). Bojar et al. (2018); Dabre et al. (2020) have demonstrated that training with multilingual data enhances the translation of low-resource languages. Related MMT research

can be roughly divided into two categories: (1) *multi-way translation*, which supports many-to-many translation through parameter sharing, multilingual representation learning, and custom joint training techniques (Aharoni et al., 2019; Yang et al., 2021; Pan et al., 2021; Tan et al., 2019); (2) *low-resource translation*, enhancing MMT under limited parallel corpora, monolingual data, or even unseen languages (Ranathunga et al., 2021; Neubig & Hu, 2018).

**Language-Specific Designs in MMT.** Among the rich literature on MMT, language-specific parameter sharing for multi-way translation is the most relevant one to our work. An effective parameter-sharing algorithm needs to decide how many parameters to share and how to share (Dabre et al., 2020). The typical manner is to share a fixed part and learn extra language-specific modules. For instance, Pires et al. (2023) assumes language-specific decoders and applies architecture search to determine the best composition of shared and language-specific encoder layers. Purason & Tättar (2022) advocates that combining shared, language-specific, and language-group-specific encoding layers benefits low-resource languages without harming high-resource languages. In contrast, another group of studies shares most of the translation model with lightweight language-specific modules that adaptively inject linguistic knowledge. For example, Zhang et al. (2021) inserts conditional language-specific (CLSR) layers in each encoder and decoder block, with a binary gate function that collects hidden representations from the shared and specialized part. Lin et al. (2021) learns a sub-network for each translation direction with shared parameters.

**Sparse Mixture of Experts (SMoE).** The concept of mixture-of-experts (MoE) can be traced back several decades (Jacobs et al., 1991; Jordan & Jacobs, 1994). It contains a series of network submodules that are utilized conditional on the input samples. The Sparsely-gated Mixture-of-Experts (SMoE) is an efficient variant of MoE, which only activates a few expert networks for each input, allowing a significant amount of increase in model parameter counts yet with minimal extra computing overheads (Shazeer et al., 2017). Numerous successes of plugging SMoE into transformer-based language models have been demonstrated in diverse NLP and computer vision applications (Fedus et al., 2022; Shazeer et al., 2017; Lepikhin et al., 2021; Fedus et al., 2021; Zuo et al., 2022; Jiang et al., 2021; Zoph et al., 2022; Riquelme et al., 2021; Yang et al., 2019).

**Routing Designs and Expert Capacity in SMoE.** Routing policy is one of the major components of SMoE, which plays an essential role in its achievable performance. Various design options are introduced to pursue an improved allocation of experts for each input sample. The classic one is a learnable router network that selects the top-$k$ experts given an input token (Lepikhin et al., 2021; Fedus et al., 2021). However, it suffers from the routing imbalance issue. Many techniques are designed to promote balanced expert assignments: injecting Gaussian noise into router networks (Shazeer et al., 2017); adding an auxiliary balancing loss to regularize routing (Lepikhin et al., 2021; Fedus et al., 2021); solving routing as a linear assignment problem (Lewis et al., 2021); using reinforcement learners (Clark et al., 2022); routing top-$k$ input tokens to each expert instead of choosing top experts per token (Zhou et al., 2022); or directly replacing learnable gates with random routing (Zuo et al., 2022; Chen et al., 2023b; Roller et al., 2021). A group of studies pioneer input-specific routing. Some studies language-specific routing (Kudugunta et al., 2021), others investigate routing with input domain information (Gururangan et al., 2022; Li et al., 2022). Linguistic characteristics like its hierarchy remain underexplored. While most studies train SMoE model with a fixed top-$k$ experts, some recent designs propose to change the expert capacity during training to adapt to multitask or lifelong learning scenarios (Chen et al., 2023c;a).

## 3 METHODOLOGY

### 3.1 PRELIMINARIES AND NOTATIONS

**Multilingual Machine Translation.** MMT is formulated as a sequence-to-sequence task, where a source language sequence is fed into an encoder, and the target language sequence will be generated from a decoder conditioned on the encoder output (Sutskever et al., 2014). The translation objective $\mathcal{L}_{\mathrm{MT}}$ is adopted to maximize the probability of the generated sequence in the target language given the source sequence. In our case, Transformer-Base (Vaswani et al., 2017) is used as our dense baseline, and our approaches are established on top of it by inserting well-designed SMoE layers.

**Sparse Mixture-of-Experts (SMoE).** In our design, we replace every other transformer block with an SMoE block for both the encoder and decoder, following the default configuration in Lepikhin et al. (2021). The SMoE block consists of $n$ experts $\{\mathtt{E}_1, \cdots, \mathtt{E}_n\}$ that are feed-forward

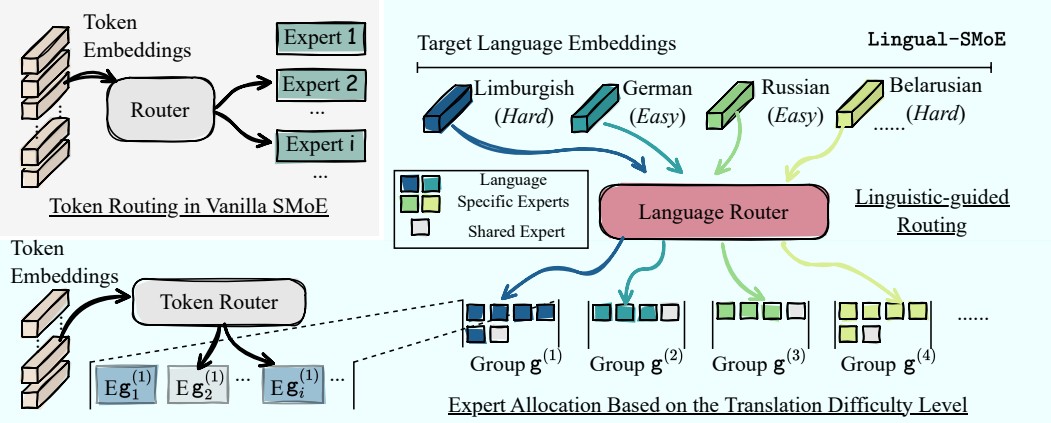

Figure 2: The overview of our proposed **Lingual-SMoE**. (■) For the vanilla SMoE, routers take each token as input and select top-$k$ experts for the following execution. (■) A hierarchical routing design is adopted in Lingual-SMoE. The linguistic-guided router **first** selects a group of experts for each target language. Note that the group size is varied based on the difficulty level of transla-tion.**Then**, another token router allocates experts from the language-specific expert group.

networks. Given an input embedding $x$, it is fed into a router network $\mathcal{G}(\cdot)$ and assigned to the most relevant experts for further processing, as shown in Figure 2 (■). The dominant design of router networks in the literature is a fully connected layer, as described below:

$$\mathcal{G} = \texttt{top-k}(\texttt{softmax}(\texttt{W}_g x)) \tag{1}$$

where $\texttt{W}_g$ are tunable parameters and $\texttt{top-k}(\cdot)$ is a selection function that outputs the largest $k$ values. The final output of an SMoE block will be a weight summarization of the features from activated experts, *i.e.*, $\sum_i^{|\mathcal{S}|} \mathcal{G}_i \cdot \texttt{E}_i(x)$. $\mathcal{S}$ denotes the index set of experts selected by the routing policy. Usually, to encourage a more uniform routing decision, an auxiliary load balancing loss $\mathcal{L}_g$ (Lepikhin et al., 2021; Shazeer et al., 2017) will be adopted for SMoE training.

## 3.2 LINGUAL-SMoE - EQUIPPING SMoE WITH LANGUAGE-GUIDED ROUTING

In this section, we detail our proposal, *i.e.*, Lingual-SMoE. As shown in Figure 2 (■), it consists of two main components (1) linguistic-guided routing (LGR) and dynamic expert allocation (DEA).

**Linguistics-Guided Routing (LGR).** We start from a pilot investigation to see whether a vanilla SMoE model naturally captures similar routing patterns for closely related languages, *e.g.,* languages from the same linguistic family. Specifically, we train a top-2 routing SMoE with the language-based routing policy, following the default configurations in Kudugunta et al. (2021). A subset of 16 lan-guages from OPUS-100 dataset is used for training and evaluation. From these translation pairs, we choose 8 languages from three different language families, *i.e.*, {{bg, sk, sl, hr}, {nb, de}, {as, mr}}, for visualizations. The first four languages (bg∼hr) belong to the Slavic language group, while nb and de are of Germanic origin, and as and mr fall into the Indo-Iranian category. To measure the similarities between routing decisions, we compute the cosine distance among different routing outputs of the corresponding en-xx (xx∈{bg∼mr}) translation directions, in the last de-coder SMoE layer. Results in Figure 3 tell us that *the routing choices show neither differentiability across language groups nor similarity between languages within the same group*. It implies that the vanilla language-based routing cannot learn the desired linguistic knowledge.

To fill in the research gap, we design a hierarchical routing policy with two-level router networks, guided by linguistic priors. In detail, for each input sequence: ① *Extracting language embedding.* We feed the target language ID into one embedding layer and two fully connected layers to produce a 512-dimensional language embedding. ② *Language routing at the first level.* In each SMoE layer, a language router $\mathcal{G}_l$ takes the language embedding as input and outputs a language-dependent expert vector. Then, $\texttt{top-k}_l(\cdot)$ function is applied on top of it to narrow down all experts to language-specific candidate experts as $\mathcal{S}_l$. ③ *Token routing at the second level.* Lastly, we allocate experts from $\mathcal{S}_l$ at the token level, to generate the final activated expert set $\mathcal{S}$. In summary, our routing policy is executed as $\sum_i^{|\mathcal{S}_l|} \sum_j^{|\mathcal{S}|} \mathcal{G}_{l,i} \cdot \mathcal{G}_j \cdot \texttt{E}_{i,j}(x)$, where $x$ is the input sample.

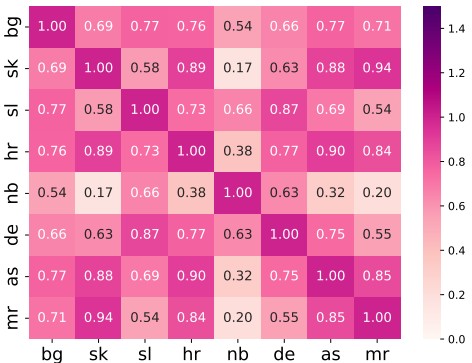

Figure 3: Routing similarities of the last decoder layer through the language routing. Three groups of target languages {bg, sk, sl, hr}, {nb, de}, {as, mr} are presented. Darker blocks imply higher similarity.

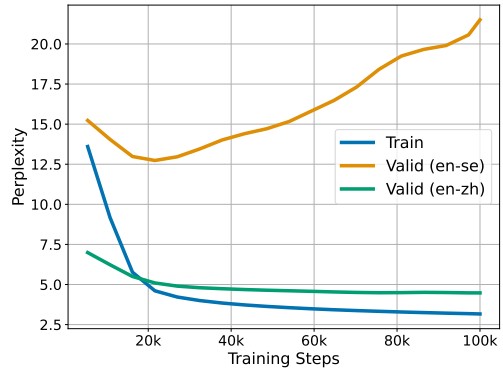

Figure 4: Train and validation perplexity of different language pairs in vanilla SMoE models. The overfitting issue is more pronounced in the low-resource pair (en-se) than in the high-resource one (en-zh).

To promote similar routing decisions between closer languages, a 2-step language grouping is proposed in our Lingual-SMoE. First, we train the embedding layer and two fully connected layers that convert a target language ID to a embedding with all target language samples. We use a contrastive loss on the language families classes to maximize the margin of embeddings of different families, [2]. Second, at the follow-up training phase, a language grouping loss $\mathcal{L}_l$ is added to the above translation, which encourages the expert allocation to be specialized in a particular language family or group. In each forward process, we compute the contrastive loss using the language routers' output. The distance measure is a cosine similarity. More details about the language grouping are included in Appendix A1 and A9. The final objective function $\mathcal{L}$ is depicted below:

$$\mathcal{L} = \mathcal{L}_{\text{MT}} + c_1 \times \mathcal{L}_g + c_2 \times \mathcal{L}_l, \tag{2}$$

where $c_1$ and $c_2$ are the hyperparameters to control the regularization effects from the load balancing loss $\mathcal{L}_g$ and language grouping loss $\mathcal{L}_l$, respectively. In our experiments, $c_1$ and $c_2$ are set to 0.05, which is determined by a grid search.

**Dynamic Expert Allocation (DEA).** If SMoE translates multiple languages with diverse complexities and a fixed model size, potential over- or under-fitting happens due to excessive or insufficient model capacity in simple or complicated scenarios respectively. It will be amplified by data imbalance. For example, as shown in Elbayad et al. (2023), SMoE models are prone to over-fitting on low-resource tasks, *i.e.* languages, or translation directions with less training data in the case of machine translation. To provide a clear picture of this severe problem, we visualize the validation perplexity of high-resource (en-zh) and low-resource (en-se) language translation, as presented in Figure 4. The results are collected from a vanilla SMoE model for MMT. It shows that *the validation perplexity of the high-resource direction decreases continuously, but the low-resource translation direction strongly overfits after 20K steps.*

---

**Algorithm 1** DEA in our proposed Lingual-SMoE.

1: **Input:** The language subset index $j$, a validation set $\mathcal{D}_{\text{val}}^j$ of the subset $j$, the number of experts per language $k_l$, a metric function $\mathcal{P}$, and a expert number growing threshold $\lambda$, number of updates $N$, maximum updates $N_{\max}$, the ratio of expert number exploring updates $r$.
2: **for** each language subset $j$ **do**
3:     Initial an indicator Improved as True;
4:     Initial the current best metric $\mathcal{P}_{\text{val(best)}}^j \leftarrow \infty$;
5:     **while** $N \leq r \times N_{\max}$ **do**
6:         **if** $\mathcal{P}_{\text{val(best)}}^j - \mathcal{P}_{\text{val}}^j < \lambda$ for $\triangle n$ iterations **then**
7:             $k_{l,j} \leftarrow k_{l,j} + 1$; Improved $\leftarrow$ False
8:         **else**
9:             $\mathcal{P}_{\text{val(best)}}^j \leftarrow \mathcal{P}_{\text{val}}^j$; Improved $\leftarrow$ True;
10:         **end if**
11:         Continue training until the next validation.
12:     **end while**
13:     **if** not Improved **then**
14:         $k_{l,j} \leftarrow k_{l,j} - 1$; rerun $\triangle n$ iterations.
15:     **end if**
16:     Fix $k_{l,j}$; train until $N_{\max}$ iterations.
17: **end for**
18: **Output:** Lingual-SMoE with top-$k_{l,j}$ routing.

---

[2]Since Indo-European languages form a majority of the OPUS-100 (58 out of 100) dataset, we treat their subfamilies as the class label for the calculation of contrastive training loss.

To address the issue and avoid laborious manual tuning, `Lingual-SMoE` offers an adaptive algorithm to decide the exact $k_l$ for the top-$k_l$ expert assignment in the language router, as illustrated in Algorithm 1. We divides different languages into three difficulty level groups, according to the data richness. In addition, we explore grouping languages considering both grammar complexity and data availability, see Appendix A2 for more details, but decide not to include grammar complexity in the difficulty metirc because of its subjectiveness. Specifically, we split the 94 validation language pairs in `OPUS-100` into three groups based on their training data size: *high-resource* ($> 0.9$M, 45 languages), *low-resource* ($< 0.1$M, 26 languages), and *medium-resource* (other, 28 languages) (Zhang et al., 2020). At the validation step per $n$ iterations on each subset, we compute the perplexity $\mathcal{P}_{\text{val}}^j$. If $\mathcal{P}_{\text{val}}^j$ does not decrease for a certain threshold, the number of candidate experts for the associated language router will be increased by updating $k_{l,j} = k_{l,j} + 1$. After the exploration stage, if $\mathcal{P}_{\text{val}}^j$ still does not decrease, we reset this iteration with $k_{l,j} = k_{l,j} - 1$ and fixed it in the rest training.

## 4 EXPERIMENTS

### 4.1 IMPLEMENTATION DETAILS

**Datasets.** We evaluate the proposed `Lingual-SMoE` on the representative multilingual neural machine translation dataset, *i.e.*, `OPUS-100` (Zhang et al., 2020) that contains 100 languages and 94 validation and test language pairs. To testify our methodology across varying language quantities, we extract datasets comprised of 16 and 50 languages from `OPUS-100`, respectively. This results in two smaller datasets, namely `OPUS-16` and `OPUS-50`, both of which have equivalent ratios of *high*, *medium*, and *low* resource languages. The three datasets are further processed using SentencePiece (Kudo & Richardson, 2018), which sets the vocabulary size to $32,000$ for `OPUS-16` and `OPUS-50` and $64,000$ for `OPUS-100`. We attach a target language ID with the source and target language sentences to identify its translation direction. See Appendix A1 for more details.

**Models and Baselines.** We compare our method with the Transformer-Base model (as *Dense*) and its SMoE variants that have 6 encoder and decoder layers, 32 experts. The input and hidden dimensions of all feed-forward networks are 512 and 2048. Meanwhile, all considered SMoE models fall into three types. ① For the vanilla SMoE model, Switch Transformer (Fedus et al., 2021) with a top-1 token-based routing (as *ST-SMoE*) and GShard (Lepikhin et al., 2021) with a top-2 token-based routing (as *GS-SMoE*) are adopted. ② We consider three improved SMoE models that incorporate language information or parameter sharing: (1) Language-specific SMoE model with fixed routing (as *LS-SMoE*) inspired by Pires et al. (2023), assigning 2 non-overlapping experts for tokens according to their source language in the encoder and target language in the decoder; (2) Hybrid SMoE model (as *Hybrid-SMoE*) from Kudugunta et al. (2021), with a top-2 token routing in the encoder and a top-2 target language routing in the decoder side; (3) Residual SMoE model (as *Residual-SMoE*) from Elbayad et al. (2023); Rajbhandari et al. (2022); Zhang et al. (2021) that augments each SMoE layer with a shared feed-forward network through a binary gate function. Note that the shared and SMoE branches are weighted accordingly for computing the final features. ③ The third group is our proposed `Lingual-SMoE`. *LGR-SMoE* stands for an SMoE model with our linguistic-guided routing, where the first-level language router selects the top 8 experts and the second-level token router activates the top 2 sequentially. In addition, to examine an interesting combination between the residual expert (Elbayad et al., 2023; Rajbhandari et al., 2022) and our linguistic-guided routing, we organically integrate them as *LGR$^{+\text{res}}$-SMoE*. For consistency of computational cost, in *LGR$^{+\text{res}}$-SMoE*, the second-level top-2 token routing is replaced by a top-1 routing. `Lingual-SMoE` further adopts the dynamic expert allocation on top of *LGR-SMoE*.

**Training and Evaluation Details.** The training processes have 35K, 100K, and 200K iterations for `OPUS-16`, `OPUS-50`, and `OPUS-100`, respectively. With a learning rate of $5 \times 10^{-4}$, we optimize models with Adam using $(\beta_1, \beta_2, \epsilon) = (0.9, 0.98, 10^{-8})$ (Kingma & Ba, 2015). The learning rate schedule follows the Inverse Square Root with a specific number of warm-up steps set to $4,000$. A temperature-based data sampling strategy is utilized to train our models (Aharoni et al., 2019). The temperature is set to 1.5 for `OPUS-16`, and 5 for `OPUS-50` and `OPUS-100`. The dynamic expert allocation uses a value of $\triangle n$ equal to $5,000$ iterations for experiments on `OPUS-16`, `OPUS-50`, and $10,000$ iterations for `OPUS-100`. In addition, the ratio of expert number exploring updates is set to 0.8, and the threshold controlling expert capacity number $\lambda$ is 0.1 for `OPUS-16`, `OPUS-50` and 0.01 for `OPUS-100`. For the memory efficiency purpose, we employ the `fp16` in training all models (Ott et al., 2018).

Table 1: Multilingual machine translation performance on OPUS16 dataset. Average BLEU scores for each translation direction and *win-ratio* are reported. We classify languages according to data amount into three groups: *high* ($> 0.9$M), *low* ($< 0.1$M), and *medium*. We compare Lingual-SMoE and its variants $\{$*LGR-SMoE*, *LGR$^{+res}$-SMoE*$\}$ with dense and SMoE baselines $\{$*Dense*, *GS-SMoE*, *ST-SMoE*$\}$, and modified SMoE models $\{$*LS-SMoE*, *Hybrid-SMoE*, *Residual-SMoE*$\}$. The total number of experts for all SMoE models is 32. The number of activated experts for each token level router is 2 except *ST-SMoE*. The number of language-dependent candidate experts is 8. The best two performances are **bold** and underlined.

| Methods | Avg. | en-xx | | | | xx-en | | | | win-rate |
|---|---|---|---|---|---|---|---|---|---|---|
| | | Avg. | *high* | *medium* | *low* | Avg. | *high* | *medium* | *low* | |
| *Dense* | 28.79 | 26.92 | 25.37 | 39.12 | 14.78 | 30.67 | 28.81 | 39.24 | 24.21 | - |
| *GS-SMoE* | 30.29 | 28.28 | 25.55 | 40.71 | 18.97 | 32.31 | 28.77 | 41.70 | 29.25 | 77% |
| *ST-SMoE* | 31.79 | 29.68 | 26.55 | 43.04 | 20.23 | 33.89 | **29.94** | **44.00** | 30.94 | **100%** |
| *LS-SMoE* | 26.75 | 22.99 | 20.06 | 37.31 | 11.72 | 30.50 | 23.69 | 42.99 | 32.01 | 33% |
| *Hybrid-SMoE* | 28.35 | 24.23 | 24.56 | 31.70 | 13.38 | 32.48 | 29.59 | 41.74 | 27.81 | 53% |
| *Residual-SMoE* | 31.97 | 30.06 | 26.52 | 43.49 | 21.58 | 33.88 | 29.84 | 43.94 | 31.25 | **100%** |
| *LGR-SMoE* | 32.32 | 31.20 | 26.46 | **46.68** | 23.23 | 33.44 | 29.76 | 43.15 | 30.28 | 97% |
| *LGR$^{res}$-SMoE* | 32.61 | 31.42 | 27.06 | 46.25 | 23.29 | 33.79 | 29.70 | 43.52 | 31.74 | **100%** |
| Lingual-SMoE | **32.71** | **31.46** | **27.11** | 46.24 | **23.34** | **33.96** | 29.87 | 43.49 | **32.02** | **100%** |

To assess all algorithms, we calculate the BLEU scores on test sets via Sacre-BLEU[3] (Post, 2018). The scores encompass the average ratings across all language pairs, such as English-to-Any (en-xx), and Any-to-English (xx-en) on the OPUS-100 dataset. Furthermore, we exhibit the *win rate*, indicating the fraction of language pairs where a method outperforms its dense counterpart (Zhang et al., 2020). Experiments are conducted using Fairseq (Ott et al., 2019) with 8 RTX A6000 GPUs.

## 4.2 LINGUAL-SMoE IMPROVES MULTI-LINGUAL MACHINE TRANSLATION

**Comparisons with Previous State-of-the-art (SoTA) Approaches.** We compare our proposed Lingual-SMoE with baselines and existing SoTA SMoE algorithms for MMT. Specifically, all models are trained and assessed on the OPUS-16 dataset with Transformer-Base as the backbone architecture. The results are summarized in Table 1. Several observations can be drawn:

▷ ① Lingual-SMoE outperforms the *Dense* and vanilla SMoE multilingual machine translation baselines with a distinct advantage. Specifically, Linguistics-Guided Routing (*LGR-SMoE*) alone achieves $\{3.53\%, 4.28\%, 2.77\%\}$, $\{2.03\%, 2.92\%, 1.13\%\}$ improvements in BLEU scores for $\{$All (Avg.), English to Any (en-xx), Any to English (xx-en)$\}$ translation directions for *Dense* and *GS-SMoE*, respectively. This clear advantage confirms the effectiveness of introducing linguistic knowledge into the routing of SMoEs. Furthermore, the English-to-Any translation direction exhibits a more notable improvement compared to the Any-to-English direction. Note that the former has diverse target language options and the latter has a single target language option, *i.e.*, English. It implies that *the language router performs a superior expert assignment when it receives diverse target language representations*. Similar observations can be found in the comparison between *LGR-SMoE* and *ST-SMoE*. *LGR-SMoE* surpasses *ST-SMoE* by a clear performance margin in terms of $\{$Avg., en-xx$\}$ translation, while has a comparable result in the case of $\{$xx-en$\}$.

▷ ② Lingual-SMoE consistently surpasses other modified SMoE models (*i.e.*, *LS-SMoE*, *Hybrid-SMoE*, and *Residual-SMoE*), demonstrating the advantage of language-guided routing in balancing the competition between shared and language-specific parameters and improving previous substandard routing. More detailed analyses lie as follows. (1) The *LS-SMoE* model trains experts separately with different language samples, resulting in nearly $2\%$ lower average BLEU score than the dense counterpart, especially when translating into languages besides English, which is about $4\%$ lower. A possible reason is that sentence pairs where English is the target language, are the majority in the datasets, which leads to a highly imbalanced training of *LS-SMoE*. In other words, the English-specific parameters will receive much more attention compared to the rest languages. In contrast, Lingual-SMoE approaches language-specific designs in a softer way by identifying and pre-routing a set of experts for each language. The language-specific expert sets are allowed to be overlapped among different languages. (2) Similarly, *Hybrid-SMoE* also suffers from imbalanced

---

[3]BLEU Signature: nrefs:1 | case:mixed | eff:no | tok:13a | smooth:exp | version:2.3.1

Table 2: Multilingual machine translation performance on OPUS50 and OPUS-100 dataset. Average BLEU scores for each translation direction and *win-rate* are reported. We compare Lingual-SMoE and its variants $\{LGR\text{-}SMoE, LGR^{+res}\text{-}SMoE\}$ with $\{Dense, GS\text{-}SMoE\}$. The best two performances are **bold** and underlined.

| Methods | Datasets | Avg. | en-xx | | | | xx-en | | | | win-rate |
| --- | --- | --- | --- | --- | --- | --- | --- | --- | --- | --- | --- |
| | | | Avg. | high | medium | low | Avg. | high | medium | low | |
| *Dense* | | 24.09 | 21.06 | 17.48 | 25.60 | 23.02 | 27.12 | 22.88 | 30.81 | 31.26 | - |
| *GS-SMoE* | | 26.45 | 23.15 | 19.23 | 29.17 | 24.14 | 29.76 | 26.16 | 33.61 | 32.49 | 83% |
| *LGR-SMoE* | OPUS50 | 27.75 | 26.19 | 22.05 | **32.92** | 26.84 | 29.31 | 26.27 | **33.76** | 30.34 | 92% |
| *LGR^res-SMoE* | | 27.31 | 25.38 | 21.59 | 32.14 | 25.30 | 29.25 | 25.71 | 33.27 | **31.69** | 94% |
| Lingual-SMoE | | **27.84** | **26.21** | **22.09** | 32.90 | **26.86** | 29.46 | 26.28 | 33.75 | 30.91 | **95%** |
| *Dense* | | 22.21 | 19.03 | 16.51 | 22.01 | 20.68 | 25.39 | 22.83 | 27.82 | 27.76 | - |
| *GS-SMoE* | | 24.82 | 20.85 | 16.88 | 24.66 | 23.99 | 28.78 | 26.73 | 32.11 | 28.81 | 79% |
| *LGR-SMoE* | OPUS100 | 27.50 | 25.58 | 22.88 | **30.66** | 24.80 | 29.42 | 27.99 | 32.45 | 28.59 | 94% |
| *LGR^res-SMoE* | | 27.44 | 25.49 | 22.68 | 30.35 | **25.18** | 29.38 | 27.77 | 32.32 | 29.00 | 95% |
| Lingual-SMoE | | **27.67** | 25.70 | 22.93 | 30.66 | 25.10 | 29.65 | 28.23 | 32.47 | 29.05 | 97% |

training even though it employs a routing mechanism since routing only by language imposes a rigid constraint on expert selection, which is mitigated by our hierarchical routing from Lingual-SMoE. (3) *Residual-SMoE* outperforms *Dense* and vanilla SMoE baseline, but its fixed routing is inferior compared to the flexibility of our hierarchical routing in Lingual-SMoE.

▷ ③ Inspired by *Residual-SMoE*, we further combine a shared expert with top-1 linguistic-guided routing ($LGR^{+res}$-*SMoE*). On OPUS-16, it enhances the performance of *LGR-SMoE* by around $0.3\%$ and increases the *win-rate* over the dense baseline to $100\%$. Extra validations about whether the fixed shared expert is necessary are presented in Table 2.

▷ ④ With Dynamic Expert Allocation, Lingual-SMoE automatically adjusts the appropriate network capacity to resolve language translations with varied complexity, by activating additional amounts of model parameters. It improves average performance on BLEU score by about $0.4\%$ over fixed *LGR-SMoE*. We see that DEA is particularly helpful for low-resource scenarios, providing an over $2\%$ increase on low-resource xx-en directions, which is consistent with our intuition.

**Evaluation across Different Number of Languages.** To examine whether Lingual-SMoE retains its advantage on datasets with more language pairs, we choose *Dense*, SMoE baselines, and better-performing methods, and train them on OPUS-50 and OPUS-100. The results are recorded in Table 2. ① We found that Lingual-SMoE continues to reach the best, outscoring the dense baseline by $\sim 5\%$ BLUE scores on OPUS-100. It further verifies the effectiveness of our adaptive language-guided routing. ② Meantime, while the $LGR^{+res}$-*SMoE* model outperforms linguistics-guided routing in small datasets, as the number of languages increases, *LGR-SMoE* overtakes its residual counterpart at most cases. A possible explanation is that a single shared residual expert starts to be insufficient to capture increased common language knowledge when the number of translation directions keeps boosting.

### 4.3 ABLATION STUDY AND EXTRA INVESTIGATION.

In this section, we further conduct an in-depth analysis of Lingual-SMoE, regarding: *i*) the contributions of its various components, *ii*) language router designs, *iii*) the number of experts and their specialization. All experiments are carried out on *OPUS-16* with the same training configurations.

**Contribution of Different Components in Linguistics-Guided Routing.** Linguistics-guided routing is divided into two steps: (1) training the language representation module and (2) training the translation model with language grouping loss and language representation initialization. Therefore, we train and evaluate models without both or without one of them, as shown in Table 3. We see that language grouping loss (Loss) is more beneficial than language representation learning (Emb). An organic combination leads to the best performance. It again confirms that linguistic guidance helps SMoE to reach better translation performance.

Table 3: Ablation on (1) language embeddings initialization (Emb) and (2) language grouping loss (Loss) in Lingual-SMoE.

| Emb | Loss | Avg. | en-xx | xx-en |
| --- | --- | --- | --- | --- |
| ✗ | ✗ | 29.18 | 27.47 | 30.90 |
| ✓ | ✗ | 30.13 | 29.35 | 30.91 |
| ✗ | ✓ | 32.10 | 31.17 | 33.03 |
| ✓ | ✓ | **32.32** | **31.20** | **33.44** |

**Comparison among Language Router Designs.** In addition to the linguistic-guided router (*learned*), we testify two other routing designs of: (1) *fixed*, the first-level router always assigns

8 fixed experts based on the target language for each input sample; (2) *random*, the first-level router randomly selects 8 experts for each input sample. The results are recorded in Table 4. Our routing policy achieves a superior performance, compared to its linguistic-agnostic counterparts. This highlights the need of appropriate sharing and specialization of model parameters for different languages in the multilingual machine translation.

**Correlation between # Allocated Experts and Linguistic Difficulty.** To understand the working mechanism of dynamic expert allocation, we track the number of selected experts for each complexity level $\in \{high, med, low\}$ in training `Lingual-SMoE`, as collected in Figure 5. We observe that along with the training, all levels gradually incorporate additional experts. Notably, the *high*-resource group witnesses the most significant surge in the number of experts, increasing from

Table 4: Ablation studies on language router designs of `Lingual-SMoE`. All use fixed expert capacity $\mathcal{S}_l = 8$.

| Method | Avg. | en-xx | xx-en |
|---|---|---|---|
| *fixed* | 16.38 | 20.94 | 11.82 |
| *random* | 28.15 | 26.49 | 29.81 |
| *learned* | **32.32** | **31.20** | **33**.44 |

8 to 11, while that of *low*-resource group allocated expert remains 8. It is within our expectations since the *high*-resource case requires a larger network capacity to process more samples, whereas the *low*-resource case needs a smaller number of experts. Dynamic Expert Allocation can be effective mainly because it expands the exploration space of SMoE by enlarging # expert candidates $|\mathcal{S}_l|$ for each language $l$. Note that there is no additional computing cost since it still only activates two experts at the second-level token routing.

**Different Number of Language-Specific Experts.** The quantity of expert candidates at the first-level language routing is a crucial hyper-parameter in our SMoE due to its significant impact on the exploration scope of the routers. The ablation results of `Lingual-SMoE` with $\{4, 8, 16, 32\}$ expert capacities in the language router are presented in Table 5. If the value of $\mathcal{S}_l$ is set to 32 (*i.e.*, using full experts), it degrades to a classic top-2 routing. `Lingual-SMoE` with $\mathcal{S}_l = 8$ seems to be a "sweet point" for superior results.

Figure 5: The dynamics of # expert.

**Linguistics-Guided Routing Visualization.** To investigate whether our routing decisions are grouped based on language similarity and vice versa, we visualize the expert assignments for different languages of the final encoder (Figure A7) and decoder (Figure A8) layers. We visualize routing decisions of three language families: Slavic languages {bg, sk, sl, hr}, Germanic languages {nb, de}, and Indo-Iranian languages {as, mr}. As shown in the

Table 5: Ablation on # language-specific experts ($\mathcal{S}_l$) in `Lingual-SMoE`. The SMoE baseline Top-2 is equivalent to the one of `Lingual-SMoE` with $\mathcal{S}_l = 32$.

| Method | Avg. | en-xx | xx-en |
|---|---|---|---|
| Top-2 (w. $\mathcal{S}_l = 32$) | 30.29 | 28.28 | 32.31 |
| *LGR* w. $\mathcal{S}_l = 16$ | 30.21 | 28.87 | 31.55 |
| *LGR* w. $\mathcal{S}_l = 8$ | **32.32** | **31.20** | **33**.44 |
| *LGR* w. $\mathcal{S}_l = 4$ | 30.24 | 29.03 | 31.45 |

visualizations, the expert distributions within each group, *i.e.*, the heatmap color patterns, are similar but not identical. For example, the Slavic language group ({bg $\sim$ hr} from the first to the fourth row) prefers experts 9 and 10 in the encoder and experts 0, 10, and 11 in the decoder. It evidences that our proposals indeed capture the linguistic hierarchy in the routing, where both language and language family types affect each token's final expert selection.

## 5 CONCLUSIONS

Sparse Mixture-of-Experts (SMoE) is a practical approach for multilingual machine translation as it allows a significant model capacity scaling while minimizing the extra computational overhead. Nevertheless, current practices overlook the linguistic characteristics that languages are hierarchically grouped and differ in complexity. In this work, we introduce a novel SMoE design for multilingual machine translation, named `Lingual-SMoE`. Our approach incorporates linguistic information into the routing process using a hierarchical router at both language and token levels. Additionally, we propose a flexible expert allocation mechanism that adjusts the number of candidate experts based on training dynamics and conditional on the translation difficulty. Numerous studies on various dense and SMoE architectures consistently showcase the performance improvements from our framework. Future plans include the extension to multiple modality scenarios.

## REPRODUCIBILITY STATEMENT

The authors have devoted a considerable amount of effort to ensure the methods and results in this paper are reproductive. Section 4.1 and Appendix A1, A9 provide details about the datasets and preprocessing. Section 4.1 guides the readers through the experimental procedure and evaluation metrics. The implementation of `Lingual-SMoE`, along with the Dense and SMoE baselines are presented in Section 4.1 as well. In addition, the codes to train and evaluate our methods are included in supplementary materials.

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

## A1 MORE IMPLEMENTATION DETAILS

Table A6: The statistics of the `OPUS-100` datasets and its sub-datasets.

| Datasets | Groups | Languages | | | | Train | Validation | Test |
|---|---|---|---|---|---|---|---|---|
| | | All | high | med | low | | | |
| OPUS-16 | 9 | 16 | 8 | 4 | 4 | $17,559,950$ | $30 \times 1000$ | $30 \times 1000$ |
| OPUS-50 | 17 | 50 | 24 | 13 | 13 | $54,444,772$ | $96 \times 1000$ | $96 \times 1000$ |
| OPUS-100 | 26 | 100 | 45 | 28 | 21 | $107,924,846$ | $188 \times 1000$ | $188 \times 1000$ |

**Language Grouping Loss Details.** Given the list of all language embeddings $\{\boldsymbol{x}_i\}$ as input, each has a corresponding language family label $y_i$. The objective of the language grouping loss $\mathcal{L}_l$ is to minimize the embedding distances for language embedding pairs of the same class, while maximizing those for pairs of different classes. The computation of $\mathcal{L}_l$ is illustrated below:

$$s_{i,j} = \frac{\boldsymbol{x}_i \cdot \boldsymbol{x}_j}{\|\boldsymbol{x}_i\|_2 \|\boldsymbol{x}_j\|_2} \tag{3}$$

$$\mathcal{L}_l(\boldsymbol{x}_i, \boldsymbol{x}_j, s) = \mathbb{1}\left[y_i = y_j\right] \|1 - s_{i,j}\| + \mathbb{1}\left[y_i \neq y_j\right] \|s_{i,j}\| \tag{4}$$

For each pair of language embeddings $\boldsymbol{x}_i$ and $\boldsymbol{x}_j$, their cosine similarity $s_{i,j}$ is computed as a measure of the embedding distance. Then the language grouping loss $\mathcal{L}_l$ of this embedding pair is $1 - s_{i,j}$ if they belong to the same language family, and $s_{i,j}$ if they are from different language families.

Table A7: Comparison of the routing mechanism of SMoE baseline models and variants of `Lingual-SMoE`. Routing Granularity: router input in encoder and decoder SMoE layer. Top-$k$: number of router activated experts. Router: whether the router is learnable or fixed. LGR: whether to enable Language Guided Routing. Shared: whether to enable a shared expert.

| Model | Routing Granularity | | Top-$k$ | | Router | LGR | Shared |
|---|---|---|---|---|---|---|---|
| | Encoder | Decoder | Language | Token | | | |
| *GS-SMoE* | Token | Token | None | 2 | Learnable | ✗ | ✗ |
| *ST-SMoE* | Token | Token | None | 1 | Learnable | ✗ | ✗ |
| *LS-SMoE* | Source | Target | None | 2 | Fixed | ✗ | ✗ |
| *Hybrid-SMoE* | Token | Target | None | 2 | Learnable | ✗ | ✗ |
| *Residual-SMoE* | Token | Token | None | 2 | Learnable | ✗ | ✔ |
| *LGR-SMoE* | Target | Target | 8 | 2 | Learnable | ✔ | ✗ |
| *LGR$^{res}$-SMoE* | Target | Target | 8 | 2 | Learnable | ✔ | ✔ |
| Lingual-SMoE | Target | Target | Dynamic | 2 | Learnable | ✔ | ✗ |

**Model Efficiency Details.** The model size and the number of tera floating point operations (TFLOPs) are reported to measure the computational cost. The TFLOPs are evaluated on a set of 128 identical samples in the OPUS dataset, with an input sequence length of 31 and a target sequence length of 25. For inference efficiency, we report average tokens processed per second (token/s) on the same test set. For training efficiency, we report the average second cost per step (s/step). We report the model efficiency metrics in Table2 of our `Lingual-SMoE` on top of one of the current SOTA SMoE models *GS-SMoE*. As shown in Table A8, Our design improves translation performance with only marginal additional parameters.

Table A8: Model efficiency of `Lingual-SMoE` and *GS-SMoE*.

| Model | Model Size | TFLOPs | Inference token/s | Training s/step |
|---|---|---|---|---|
| *GS-SMoE* | 148.8M | 1.05 | 469.84 | 1.208 |
| Lingual-SMoE | 149.48M | 1.05 | 456.83 | 1.205 |

## A2 MORE EXPERIMENT RESULTS

**Extra Evaluations.** We provide the ChrF, COMET, ROUGE-L, and METEOR scores of *Dense*, *GS-SMoE*, and *Lingual-SMoE* trained on OPUS-50 datasets in Table A10, which show that the advantages of our *Lingual-SMoE* across different metrics. The ChrF score is computed using Sacre-BLEU. The COMET score is calculated using the COMET framework. The ROUGE-L, and ME-TEOR scores are calculated with the HuggingFace evaluate library.

Table A9: All languages in `OPUS-100` and their corresponding abbreviations (abbr.) and language groups.

| Language | abbr. | Group | Language | abbr. | Group |
|---|---|---|---|---|---|
| Hebrew | he | afroasiatic | Portuguese | pt | indo-european romance |
| Arabic | ar | afroasiatic | Romanian | ro | indo-european romance |
| Maltese | mt | afroasiatic | Spanish | es | indo-european romance |
| Hausa | ha | afroasiatic | French | fr | indo-european romance |
| Amharic | am | afroasiatic | Italian | it | indo-european romance |
| Vietnamese | vi | austroasiatic | Catalan | ca | indo-european romance |
| Khmer | km | austroasiatic | Galician | gl | indo-european romance |
| Malay | ms | austroasiatic | Walloon | wa | indo-european romance |
| Indonesian | id | austroasiatic | Occitan | oc | indo-european romance |
| Malagasy | mg | austroasiatic | Aragonese | an | indo-european romance |
| Mongolian | mn | mongolic | Bulgarian | bg | indo-european slavic |
| Sinhala | si | dravidian | Slovak | sk | indo-european slavic |
| Malayalam | ml | dravidian | Slovenian | sl | indo-european slavic |
| Tamil | ta | dravidian | Croatian | hr | indo-european slavic |
| Telugu | te | dravidian | Polish | pl | indo-european slavic |
| Kannada | kn | dravidian | Ukrainian | uk | indo-european slavic |
| Lithuanian | lt | indo-european baltic | Russian | ru | indo-european slavic |
| Latvian | lv | indo-european baltic | Bosnian | bs | indo-european slavic |
| Irish | ga | indo-european celtic | Serbian | sr | indo-european slavic |
| Welsh | cy | indo-european celtic | Czech | cs | indo-european slavic |
| Breton | br | indo-european celtic | Macedonian | mk | indo-european slavic |
| Scottish Gaelic | gd | indo-european celtic | Serbo-Croatian | sh | indo-european slavic |
| German | de | indo-european germanic | Belarusian | be | indo-european slavic |
| Danish | da | indo-european germanic | Basque | eu | isolate |
| Dutch | nl | indo-european germanic | Japanese | ja | japonic |
| English | en | indo-european germanic | Georgian | ka | kartvelian |
| Swedish | sv | indo-european germanic | Korean | ko | koreanic |
| Icelandic | is | indo-european germanic | Kinyarwanda | rw | niger-congo |
| Norwegian | no | indo-european germanic | Xhosa | xh | niger-congo |
| Norwegian Bokmal | nb | indo-european germanic | Igbo | ig | niger-congo |
| Afrikaans | af | indo-european germanic | Zulu | zu | niger-congo |
| Norwegian Nynorsk | nn | indo-european germanic | Yoruba | yo | niger-congo |
| Western Frisian | fy | indo-european germanic | Chinese | zh | sino-tibetan |
| Yiddish | yi | indo-european germanic | Burmese | my | sino-tibetan |
| Limburgish | li | indo-european germanic | Thai | th | tai-kadai |
| Dzongkha | dz | nilo-saharan | Turkish | tr | turkic |
| Persian | fa | indo-european indo-iranian | Azerbaijani | az | turkic |
| Bangla | bn | indo-european indo-iranian | Uzbek | uz | turkic |
| Assamese | as | indo-european indo-iranian | Uyghur | ug | turkic |
| Gujarati | gu | indo-european indo-iranian | Kyrgyz | ky | turkic |
| Tajik | tg | indo-european indo-iranian | Kazakh | kk | turkic |
| Nepali | ne | indo-european indo-iranian | Tatar | tt | turkic |
| Punjabi | pa | indo-european indo-iranian | Turkmen | tk | turkic |
| Urdu | ur | indo-european indo-iranian | Hungarian | hu | uralic |
| Hindi | hi | indo-european indo-iranian | Estonian | et | uralic |
| Marathi | mr | indo-european indo-iranian | Finnish | fi | uralic |
| Pashto | ps | indo-european indo-iranian | Northern Sami | se | uralic |
| Kurdish | ku | indo-european indo-iranian | Esperanto | eo | constructed |
| Odia | or | indo-european indo-iranian | Greek | el | indo-european hellenic |
| Armenian | hy | indo-european armenian | Albanian | sq | indo-european albanian |

Table A10: Multilingual machine translation performance on `OPUS50` dataset.

| Model | ChrF | COMET | ROUGE-L | METEOR |
|---|---|---|---|---|
| *Dense* | 43.14 | 72.82 | 40.99 | 41.47 |
| *GS-SMoE* | 45.36 | 74.46 | 42.38 | 43.15 |
| *Lingual-SMoE* | **46.56** | **75.52** | **43.06** | **43.96** |

**Dynamic Expert Allocation with Grammar Complexity and Data Abundance.** As outlined in the introduction, our study examines language difficulty from two perspectives: grammatical complexity and resource availability, which are combined by assigning corresponding scores. (1) In terms of available data, following (Zhang et al., 2020), languages with sample sizes exceeding 0.9

Table A11: Performance comparisons of Lingual-SMoE$_{extra}$ with dynamic routing according to language difficulty defined by grammar complexity and data abundance, compared to *Dense* and vanilla SMoE *GS-SMoE*. Average BLEU scores for each direction are reported.

| Methods | Avg. | en-xx | | | | xx-en | | | |
|---|---|---|---|---|---|---|---|---|---|
| | | Avg. | *high* | *medium* | *low* | Avg. | *high* | *medium* | *low* |
| *Dense* | 28.79 | 26.92 | 25.37 | 39.12 | 14.78 | 30.67 | 28.81 | 39.24 | 24.21 |
| *GS-SMoE* | 30.29 | 28.28 | 25.55 | 40.71 | 18.97 | 32.31 | 28.77 | 41.70 | 29.25 |
| Lingual-SMoE$_{extra}$ | **32.55** | **31.36** | **27.05** | **45.82** | **23.59** | **33.73** | **29.95** | **43.61** | **30.66** |

million samples are categorized as high-resource languages, while those with less than 0.1 million samples are considered low-resource languages. Languages falling between the two thresholds are classified as medium-resource languages. Scores of 0, 1, and 2 are assigned to low, medium, and high resource languages, respectively, with higher scores indicating greater data scarcity. (2) Regarding grammatical complexity, each language is rated on a scale of 1 to 5 using GPT-4, reflecting the level of difficulty from easy to hard. (3) Then the language difficulty metric is computed by summing the scores derived from resource availability and grammatical complexity. During training, languages are categorized as easy, medium, and hard based on their language difficulty scores, maintaining the same distribution of high, medium, and low resource languages. Next, we train Lingual-SMoE$_{extra}$ with the language difficulty groups in the OPUS-16 settings and compare it to the dense and SMoE baselines. As shown in Table S1, Lingual-SMoE$_{extra}$ consistently outperforms the baseline models due to the incorporation of both grammar and data information. The reason why we exclude language difficulty when combining the grammar complexity score and the data availability score is twofold. First, the amount of data is an objective metric, but grammar complexity is relatively subjective for people with different first languages and education levels. Also, the definition of grammar complexity varies from GPT to human, and even from case to case within GPT. So we decide not to include grammar complexity in our Lingual-SMoE.

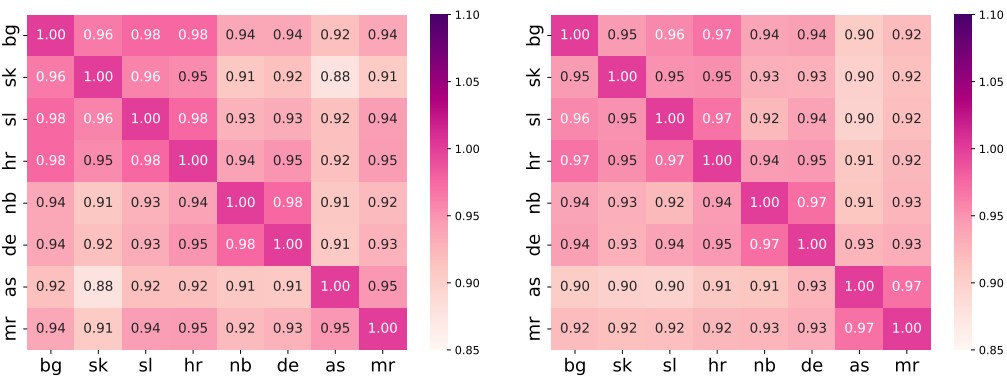

(a) Encoder Routing Decision Similarity.  (b) Decoder Routing Decision Similarity.

Figure A6: Routing decision similarities of the last encoder and decoder SMoE layer of Lingual-SMoE trained on OPUS-100 for en-xx language pairs. Three groups of target languages {bg, sk, sl, hr}, {nb, de}, {as, mr} are presented. Darker blocks imply higher similarity.

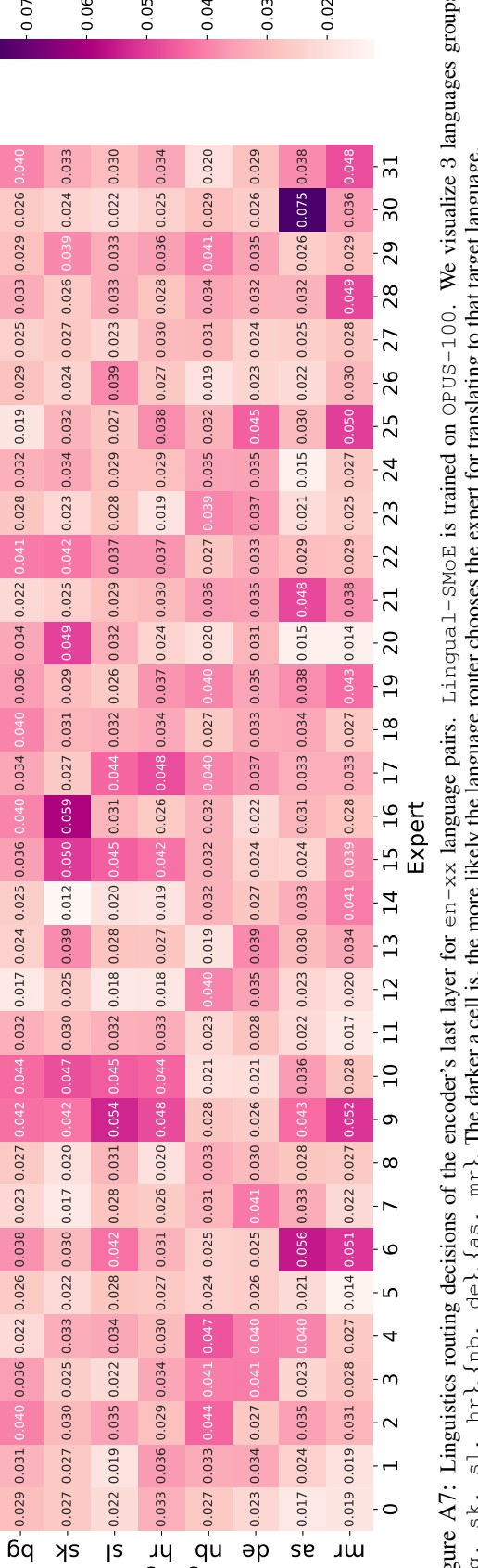

Figure A7: Linguistics routing decisions of the encoder's last layer for `en-xx` language pairs. `Lingual-SMoE` is trained on `OPUS-100`. We visualize 3 languages groups {`bg`, `sk`, `sl`, `hr`}, {`nb`, `de`}, {`as`, `mr`}. The darker a cell is, the more likely the language router chooses the expert for translating to that target language.

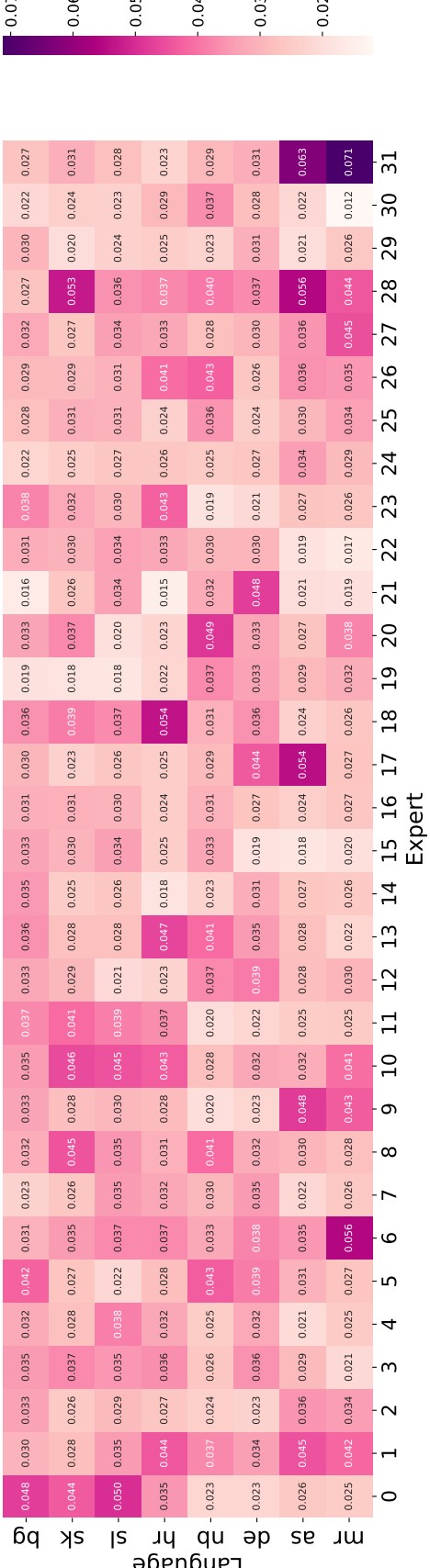

Figure A8: Routing decisions of the decoder's last layer for en-xx language pairs. Lingual-SMoE is trained on OPUS-100. We visualize 3 languages groups {bg, sk, sl, hr}, {nb, de}, {as, mr}. The darker a cell is, the more likely the language router chooses the expert for translating to that target language.

