# OpenReview forum: "Sparse MoE with Language Guided Routing for Multilingual Machine Translation"
_ICLR.cc/2024/Conference — ICLR 2024 poster_

### Official Review · Reviewer_BUYj · 2023-10-16

**Soundness:** 3 good
**Presentation:** 3 good
**Contribution:** 3 good
**Rating:** 8
**Confidence:** 4

**Summary:**

The presented work proposes to add linguistic information into the mixture-of-expert routing process and shows the effectiveness on the OPUS-100 dataset compared to other gating choices.

**Strengths:**

- Interesting work adding linguistic features and language complexity into the routing behaviour
- Innovative design of an adaptive expert allocation mechanism
- The paper is well written and easy to follow
- Nice ablations for Linguistically Guided routing (Table 3) and the Language Router Designs (Table 4)

**Weaknesses:**

### Weaknesses

- **[major]**: Please provide the `sacrebleu` hash that is used for evaluation such that scores can be reproducible
- **[major]**: While BLEU is still widely used, there are now better metrics to use for Machine Translation that correlate better with human judgment as seen in [Results of WMT22 Metrics Shared Task: Stop Using BLEU – Neural Metrics Are Better and More Robust](https://aclanthology.org/2022.wmt-1.2) (Freitag et al., WMT 2022), specifically I'd recommend including chrF and COMET scores.
- **[major]**: One of the key takeaways of [Pires et al. 2023](https://aclanthology.org/2023.acl-long.825/) is that target language specific routing is helpful in the encoder and not only in the decoder and only source language routing the encoder hinders learning. This is true for both shared as well as language-specific decoders according to their experiments (see their Table 1 + Table 5). I think the baseline (1) in the presented work could be improved with two approaches to more closely match their setup by 1) converting the source language routing in the top 25% of encoder layers to target language routing and 2) deploying their proposed dense pre-training approach.
- **[minor]**: Building on top of the previous point, I think even LGR-SMoE could benefit from the dense pre-training as this was also shown to be very beneficial in both [Pires et al. 2023](https://aclanthology.org/2023.acl-long.825/) and [Tricks for Training Sparse Translation Models](https://aclanthology.org/2022.naacl-main.244) (Dua et al., NAACL 2022).
- **[minor]**: The justification on p.7 that (1) performs much more biased towards English doesn't make much sense as the results in Table 1 only include `X` $\leftrightarrow$ `en` scores and OPUS-100 is an English-centric corpus? In any case, adding per language pair scores for all evaluation directions in the appendix would be beneficial for all methods. It might also be worth to have a LS baseline where only the decoder is language-specific for the target language and we do not have any language specific routing in the encoder since this granularity might be too much for the small english-centric OPUS-100 dataset.
- **[minor]**: How big is the inference speed / parameter count / memory consumption overhead from the additional language router? I think some concrete benchmarking numbers would be helpful here.
- **[minor]**: There are no details about the data epochs, are all of the presented models trained until convergence and is it single epoch? How was checkpoint selection done since we likely can see overfitting for some of the lower resource languages in e.g. LS-SMoE?
- **[major]**: It is unclear how the proposed approach influences the zero-shot translation quality and/or code-switched payloads.

---
### Typos & Minor Comments

- Change citation of "Nllbteam" to proper capitalization and spacing
- Double check bibliography some works such as e.g. [Massively Multilingual Neural Machine Translation](https://aclanthology.org/N19-1388) (Aharoni et al., NAACL 2019) or [Learning Language-Specific Layers for Multilingual Machine Translation](https://aclanthology.org/2023.acl-long.825) (Pires et al., ACL 2023) have published version available that should be cited.

**Questions:**

- p.6: What is $\alpha$ for the Adam algorithm? Shouldn't this be either $\beta_1$, $\beta_2$ or $\epsilon$? (see Table 4 in [Schmidt et al., 2021](https://proceedings.mlr.press/v139/schmidt21a.html)). Also, please cite the original Adam paper accordingly.
- Is the code going to be open-sourced?

---

> ### Author Response · Authors · 2023-11-20
> **Responses to Reviewer BUYj [Cons1-3]**
>
> We sincerely appreciate your careful review and a great summary of our contributions. And thank you for the constructive comments. Please see our point-by-point response below.
>
> **[CONS1 Evaluation Reproductivity]** We appreciate this constructive suggestion. The Sacrebleu Hash is as below. It has been added in the updated draft as a footnote on page 6, highlighted in orange.
> BLEU Signature: nrefs:1 | case:mixed | eff:no | tok:13a | smooth:exp | version:2.3.1
>
> **[CONS2 Evaluation on Other Metrics]**
>
> Thanks for your comments. We have provided the ChrF, COMET, ROUGE-L, and METEOR scores of $\textit{Dense}$, $\textit{GS-SMoE}$, and $\texttt{Lingual-SMoE}$ trained on OPUS-50 datasets in Table S1, which show that our $\textit{Lingual-SMoE}$ is competitive across different metrics. The ChrF score is computed using Sacre-BLEU. The COMET score is calculated using the COMET framework. The ROUGE-L, and METEOR scores are calculated with the HuggingFace evaluate library. Table S1 is included in the updated draft, in Appendix.
>
> Table S1 Multilingual machine translation performance on OPUS-50 dataset.
> | model | ChrF | COMET | ROUGE-L | METEOR |
> | --------------------- | --------- | --------- | --------- | --------- |
> | $\textit{Dense}$ | 43.14 | 72.82 | 40.99 | 41.47 |
> | $\textit{GS-SMoE}$ | 45.36 | 74.46 | 42.38 | 43.15 |
> | $\texttt{Lingual-SMoE}$ | **46.56** | **75.52** | **43.06** | **43.96** |
>
> **[CONS3 Improving LS-SMoE]**
>
> Thank you for the helpful suggestion. We understand your concerns about building a stronger baseline that better aligns with the original study. However, we decided not to include $\textit{LS-SMoE}$ with source and target language-specific routing in the encoder, and dense pre-training as baselines due to the following reasons:
> - We have previously experimented with the $\textit{LS-SMoE}$ encoder with both source and target language-specific parameters. However, the performance of such a combination is suboptimal and inferior to our current choice;
> - While dense pre-training has been verified to benefit $\textit{LS-SMoE}$, training baseline models from scratch better cope with our settings, which can exclude possible interference for examining the effect of language-guided routing, and whether it learns language-specific and language-family-specific knowledge. Therefore, $\textit{LS-SMoE}$ with dense pre-training is not included in our baselines for fairness of comparison.
>
> *[Extra experiment on $\textit{LS-SMoE}$]* We have reported the performance of the two $\textit{LS-SMoE}$ variants in Table S2. The first variant we have examined is $\textit{LS-SMoE}^{+hybrid}$, where the encoder is converted from using only source language routing to a mixture of source and target language routing (i.e., source language routing in the two lower MoE layers and target language routing in the upper). As the ratio of target-level routing increases, $\textit{LS-SMoE}^{+hybrid}$ performs better on the more diverse setting of $\texttt{en-xx}$ with an average 2.53 increase in BLEU, yet suffering from a 7.23 decrease in the BLEU score on the setting of $\texttt{xx-en}$. Another variant we studied is $\textit{LS-SMoE}^{+dense}$, whose weights are initialized from $\textit{Dense}$. More specifically, the expert feed-forward networks in $\textit{LS-SMoE}^{+dense}$ are initialized from the same feed-forward networks in $\textit{Dense}$ in the corresponding layers. The result of $\textit{LS-SMoE}^{+dense}$  is much better than $\textit{LS-SMoE}$, but still falls behind our $\textit{LGR-SMoE}$ without dense pre-training.
>
> *[Possible reasons]* The possible reason for the reverse effect of optimizing $\textit{LS-SMoE}$ model with both source and target language routing in the encoder could be attributed to the complexity of SMoE optimization by nature [1,2]. Single type routing policy is already challenging. With a fused routing policy, the difficulty level of optimization will further increase. In summary, it will be interesting to discover different routing policies of the SMoE model; however, it falls beyond the scope of our paper and we will leave it in the future.
>
> [1] Sparse MoE as the New Dropout: Scaling Dense and Self-Slimmable Transformers
>
> [2] TA-MoE: Topology-Aware Large Scale Mixture-of-Expert Training
>
> Table S2 Average BLEU score of multilingual machine translation performance on OPUS-16 dataset
> | | all | en-xx | xx-en |
> | ------------------------------------- | --------- | --------- | --------- |
> | $\textit{LS-SMoE}$ | 26.75 | 22.99 | 30.50 |
> | $\textit{LS-SMoE}^{+hybrid}$ | 24.39 | 25.52 | 23.26 |
> | $\textit{LS-SMoE}^{+dense}$ | 28.45 | 23.63 | 33.27 |
> | $\textit{LGR-SMoE}$ | **32.32** | **31.20** | **33.44** |

---

> ### Author Response · Authors · 2023-11-20
> **Responses to Reviewer BUYj [Cons4-5]**
>
> **[CONS4 Analysis of LS-SMoE]**
>
> *[Clarification on the analysis of $\textit{LS-SMoE}$]* We respectfully point out that the intention of the argument in P7 2(1) is that the fixed routing of $\textit{LS-SMoE}$ leads to an imbalanced training of experts, where experts corresponding to English are activated much more often, due to its quantitative superiority and the English-centric nature of the OPUS dataset. Specifically for the SMoE model, the results in [3,4] show that the routing decision in the decoder is more language-specific. This is also consistent with the design in LSL [5] which the decoder is completely language-specific. Thus, the effect of imbalance activation of $\textit{LS-SMoE}$ is more severe in the en-xx direction, which requires higher routing diversity to explore language-specific parameters. Regarding the score per language pair, since the number of language pairs is relatively large ($96$ for OPUS-50, $188$ for OPUS-100), we will find a better format to present the detailed result in the final version.
>
> [3] Beyond Distillation: Task-level Mixture-of-Experts for Efficient Inference
>
> [4] Memory-efficient NLLB-200: Language-specific Expert Pruning of a Massively Multilingual Machine Translation Model
>
> [5] Learning Language-Specific Layers for Multilingual Machine Translation
>
> *[Decoder language-specific SMoE baseline]* We have implemented an SMoE baseline $\textit{Hybrid-SMoE}$ where only the decoder is target language-specific routing, while the encoder is only dependent on the input token. The only difference to $\textit{LS-SMoE}$ is that the router of $\textit{Hybrid-SMoE}$ is learnable. From the results in Table 1 and Table 4, we can conclude that learnable routing is a useful technique for improving SMoE for MMT. Therefore, we consider $\textit{Hybrid-SMoE}$ as an improved version of the reviewer's proposal. The performance of $\textit{Hybrid-SMoE}$ shows that language-level routing alone is too coarse and justifies combining language, language family, and input information in a hierarchical router.
>
> **[CONS5 Implementation Details]**
>
> *[Model efficiency]* Thanks for the interesting question. We measure model efficiency by parameter size, the number of tera floating point operations (TFLOPs), and training and inference time. in Table S3. In particular, we record the average tokens processed per second (token/s) on the same test set to measure inference efficiency. For training efficiency, we report the average second cost per step (s/step). We compare our $\texttt{Lingual-SMoE}$ to one of the current SOTA SMoE models $\textit{GS-SMoE}$. The result shows that with significant improvement in translation performance, our design demands only marginal additional parameters.
>
> Table S3 Comparison of Model efficiency of $\texttt{Lingual-SMoE}$ and $\textit{GS-SMoE}$
> | model | Model Size | TFLOPs | Inference efficiency (token/s) | Training efficiency (s/step) |
> | ----------------------- | ---------- | ----------- | ------------------------------ | ---------------------------- |
> | $\textit{GS-SMoE}$ | 148.8 M | 1.05 | 469.84 | 1.208 |
> | $\texttt{Lingual-SMoE}$ | 149.48 M | 1.05 | 456.83 | 1.205 |
>
> *[Data epoch]* As for the details of the data epochs, we show the training steps and corresponding data epoch on OPUS-16, OPUS-50, OPUS-100 in Table S4. We conducted a preliminary experiment to determine the training steps of our models to ensure that they are well-trained. Specifically, we train $\textit{GS-SMoE}$ and $\textit{Dense}$ on OPUS-100 for 10 epochs (about 918k steps). We find that the losses for both models converge quickly at about 100K steps and then decrease at a much slower rate. Similar experiments are performed on all datasets so that we can finally determine the training steps that ensure model convergence.
>
> Table S4 Training step and epoch details.
> | Dataset | Training step | Epoch |
> | -------- | ------------- | ----- |
> | OPUS-16 | 35k | 6.44 |
> | OPUS-50 | 100k | 1.89 |
> | OPUS-100 | 200k | 4.35 |
>
> *[Checkpoint selection]* Since the validation and test sets are separate, we use the overall language perplexity on the validation set as the metric to select checkpoints during training. This setting applies to all methods to ensure a fair comparison.

---

> ### Author Response · Authors · 2023-11-20
> **Responses to Reviewer BUYj [Cons6-7]**
>
> **[CONS6 Zero-shot Translation Evaluation]**
> Applying the MMT model to conduct zero-shot translation is an interesting suggestion and application. Due to the limited time of the rebuttal period, we collected some non-English centric test sets from the Tatoeba challenge [6] and evaluated $\textit{Dense}$ and $\textit{Lingual-SMoE}$ on them. Although our $\textit{Lingual-SMoE}$ is not specialized for multi-to-multi translation, it outperforms its Dense counterpart, as shown in Table S5.
>
> [6] https://github.com/Helsinki-NLP/Tatoeba-Challenge
>
> Table S5 Zero-shot translation performance
> | zero-shot | de-bg | de-eu | de-hr | de-nb | sl-de | de-bg | de-eu |
> | ----------------- | ------------ | ------------- | ------------ | ------------- | -------------- | ------------ | ------------- |
> | $\textit{Dense}$ | 5.43 | 2.27 | 2.67 | 7.24 | 8.59 | 5.43 | 2.27 |
> | $\textit{LGR-SMoE}$ | **7.33** | **3.25** | **4.84** | **8.27** | **10.37** | **7.33** | **3.25** |
>
> **[CONS7 Bibliography and Citation]**
> Thanks for the detailed review. We have addressed them in our updated draft.
> - We have changed "Nllbteam" to "NLLB Team" in the bibliography.
> - We have updated the information about the two published versions of the paper.
> - We adopt the original Adam in [7]. We have added it to the bibliography and cited it in P6. The hyperparameters of Adam are changed from $\alpha$, $\beta$ to $\beta_1$, $\beta_2$, $\epsilon$, which are set to $0.9, 0.98, 10^{-8}$, respectively.
> We will check all the bibliographies to make sure the information is up to date in the final version.
>
> [7] Adam: A Method for Stochastic Optimization
>
> **[Question: Code Availability]** The core implementation codes are provided in the supplement with training and evaluation scripts. URLs for additional code will be published in the final version.

---

> > ### Comment · Reviewer_BUYj · 2023-11-20
> > **Thanks for the response!**
> >
> > Thank you for the very detailed response and additional experiments! All my weaknesses have been addressed. As my initial score was already an accept (8), I will keep the current rating as is.

---

> > > ### Author Response · Authors · 2023-11-21
> > > **Thanks to Reviewer BUYj**
> > >
> > > Thank you for your prompt reply. We are glad that we have addressed your concerns. Again, we sincerely appreciate your time and support.
> > >
> > > Best wishes,
> > >
> > > Authors

---

### Official Review · Reviewer_xU8U · 2023-10-30

**Soundness:** 4 excellent
**Presentation:** 4 excellent
**Contribution:** 3 good
**Rating:** 8
**Confidence:** 5

**Summary:**

This paper introduces a novel approach known as Lingual-SMoE, aimed at enhancing multilingual machine translation. In contrast to conventional methods, Lingual-SMoE takes into consideration the distinct linguistic features and complexities of various languages. The experimental findings are promising in specific translation directions, as Lingual-SMoE consistently outperforms traditional techniques in multilingual translation tasks. For instance, it exhibits an increase of over 5% in BLEU scores on the OPUS-100 dataset. This underscores Lingual-SMoE's capability to effectively handle translation tasks for various languages, improving overall accuracy

**Strengths:**

1.	Consideration of Linguistic Characteristics: This approach takes into account the linguistic hierarchy and the complexity of various languages, thus enhancing its performance in multilingual machine translation tasks. The initiative is highly motivated and intriguing in terms of how it organizes models within the MoE framework to match the specific characteristics of language pairs.

2.	Hierarchical Routing Strategy: In order to implement this proposed concept, it introduces a hierarchical routing strategy that takes into consideration language families and token-level information. This results in a more efficient allocation of experts, optimizing the utilization of resources and routing decisions.

3.	Adaptive Expert Allocation: Another noteworthy proposal is the mechanism for adaptive expert allocation, which can automatically adjust the capacity of experts based on the translation difficulty of each language. This helps address issues related to overfitting and underfitting. The method is both intuitive and technically robust.

4.	Empirical Validation: Extensive experiments, carried out on a variety of language pairs and scales, have consistently demonstrated the effectiveness of this approach across different data resources and language quantities. It consistently leads to a significant improvement in performance, particularly on the OPUS-100 dataset.

**Weaknesses:**

1.	In the paper, the authors claim to address the issues of Linguistic Hierarchy and Language Complexity. The former refers to the categorization of languages based on their language families, while the latter pertains to "grammar complexity" or "language difficulty." However, I believe that the second claim may not be appropriate, as it appears to relate more to the availability of training data rather than addressing the inherent challenges of the languages themselves. Additionally, there is a lack of experiments or analyses to support the claim that their method genuinely considers the intrinsic difficulty of languages.

2.	In the experiments, for the purpose of fair comparisons between the baselines and the proposed methods, it is recommended that the authors report the sizes of the models. It seems that the improvements may be attributed to the increased number of experts. Therefore, providing and discussing the size of model parameters, along with the computational complexity of the proposed hierarchical framework in terms of training and decoding time, would be beneficial.

3.	Some technical details require further clarification, and I would like to refer to the specific questions raised in the Questions.

4.	There are several typos that need correction, such as:

   - There are issues with citation formats and grammatical agreement in the last paragraph of the Related Work section.
   - In Section 4.2, "Table 1" is incorrectly cited as "Table 4.2."

**Questions:**

1.	As depicted in Figure 2, how is the difficulty of a language determined?

2.	Based on your findings in Table 1 and Table 2, it appears that your method has demonstrated improved performance in the en-xx language directions. However, the performance in the xx-en language directions appears to be less favorable than previous results, particularly in the context of medium to high-resource translation tasks. Have you conducted a detailed analysis of this phenomenon, or is it possible that your method still has limitations when it comes to xx-en tasks?

3.	Regarding Eq. (2) in Section 3, it is mentioned that \alpha and \beta are empirically set to 0.05. However, in the Implementation Details section, \alpha and \beta are set to 0.98. Could you please clarify whether these refer to the same parameters?

**Details Of Ethics Concerns:**

NIL

---

> ### Author Response · Authors · 2023-11-20
> **Responses to Reviewer xU8U [Cons1-2]**
>
> Thanks for acknowledging our methodology and experiment as meaningful and sufficient. We provide pointwise responses to your concerns about our experiments as below.
>
> **[CONS1 Dynamic Routing According to Language Difficulty]**
>
> Thanks for pointing this out. We examine language difficulty from two perspectives: grammatical complexity and resource availability, and combine them in dynamic routing. Here, we (1) report the results of the additional dynamic routing experiments that integrate data abundance and grammatical complexity; (2) explain why we consider the data abundance factor; and (3) appropriately refine paper writing to avoid potential confusion.
>
> - *[Additional experiment results]* First, we compute language difficulty scores by adding grammar difficulty (rated by GPT-4 from 1-5) and data abundance scores (0,1,2 for high/medium/low resource) as the metrics to divide languages into easy, medium, and difficult groups, following the ratio of high/medium/low resource language numbers. Then we train a $\texttt{Lingual-SMoE}^{+grammar}$ with the language difficulty groups and compare it to the dense and SMoE baselines. All models are trained in OPUS-16 settings. As shown in Table S1, $\texttt{Lingual-SMoE}^{+grammar}$ outperforms the baseline models with the incorporation of both grammar and data information.
>
> Table S1 Performance comparisons of $\texttt{Lingual-SMoE}^{+grammar}$ with dynamic routing according to language difficulty defined by grammar complexity and data abundance, compared to $\textit{Dense}$ and vanilla SMoE $\textit{GS-SMoE}$. Average BLEU scores for each direction are reported.
> | | all | en-xx | high | mid | low | xx-en | high | mid | low |
> | ------------------------------- | --------- | --------- | --------- | --------- | --------- | --------- | --------- | --------- | --------- |
> | $\textit{Dense}$ | 28.79 | 26.92 | 25.37 | 39.12 | 14.78 | 30.67 | 28.81 | 39.24 | 24.21 |
> | $\textit{GS-SMoE}$ | 30.29 | 28.28 | 25.55 | 40.71 | 18.97 | 32.31 | 28.77 | 41.70 | 29.25 |
> | $\texttt{Lingual-SMoE}^{+grammar}$ | **32.55** | **31.36** | **27.05** | **45.82** | **23.59** | **33.73** | **29.95** | **43.61** | **30.66** |
>
> - *[Reason why we mainly consider data availability]* The reason we do not include grammar complexity is: First, the data amount is an objective value, but grammar complexity is subjective for people with different first languages and education. Also, the degree of grammar complexity varies from GPT to human. So we decided not to consider grammar complexity in our $\texttt{Lingual-SMoE}$.
>
> **[CONS2 Model Efficiency]**
>
> Thanks for the comments. We report model size, the number of tera floating point operations (TFLOPs), inference, and training efficiency in Table S2. In particular, we record the average tokens processed per second (token/s) on the same test set to measure inference efficiency. For training efficiency, we report the average second cost per step (s/step). We compare our $\texttt{Lingual-SMoE}$ to one of the current SOTA SMoE models $\textit{GS-SMoE}$. With significant translation performance, our design requires only marginal additional parameters.
>
> Table S2 Comparison of Model efficiency of $\texttt{Lingual-SMoE}$ and $\textit{GS-SMoE}$
> | model | Model Size | TFLOPs | Inference efficiency (token/s) | Training efficiency (s/step) |
> | ----------------------- | ---------- | ----------- | ------------------------------ | ---------------------------- |
> | $\textit{GS-SMoE}$ | 148.8 M | 1.05 | 469.84 | 1.208 |
> | $\texttt{Lingual-SMoE}$ | 149.48 M | 1.05 | 456.83 | 1.205 |
>
> We clarify that the improvements are not driven by the increased number of experts, since all the baseline SMoE models and our designs (except $\textit{ST-SMoE}$ with top-$1$ routing) activate only the top 2 experts during the training.

---

> ### Author Response · Authors · 2023-11-20
> **Responses to Reviewer xU8U [Cons3-4]**
>
> **[CONS3 Inconsistent Improvement Across Langauges]**
>
> *[Detailed observation]* Thanks for pointing this out. (1) From Tables 1 and 2, the translation direction $\texttt{xx-en}$ benefits less from our method than $\texttt{en-xx}$ to a certain extent. (2) Taking the improvement of $\texttt{Lingual-SMoE}$ from its SMoE counterpart $\textit{GS-SMoE}$ as an example: the improvements are greater in OPUS-100 ($2.84\%$) than in OPUS-16/50 ($2.42\%$ and $1.39\%$); (3) among low/medium/high-resource language groups (in OPUS-16 as an example), the improvements are more obvious in low and medium resource languages ($3.66\%$ and $3.66\%$) compared to high resource languages ($1.33\%$).
>
> *[Detailed analysis]* Several conclusions can be drawn from the results: (1) $\texttt{xx-en}$ direction translation shows less improvement compared to $\texttt{en-xx}$. (2) Low-resource languages benefit more from LGR. (3) When the number of languages increases, the performance gap between $\texttt{en-xx}$ and $\texttt{xx-en}$ decreases, which indicates that more languages help our method improve en-xx and xx-en directions simultaneously.
>
> *[Possible reasons]* The possible explanations for these results are: (1) in $\texttt{en-xx}$, we design routing with more exploration capacity because the "$\texttt{xx}$" side covers more language families, which is the reason for the higher performance gain. (2) In low-resource scenarios, a linguistic prior drawn from closer family languages is more helpful, which is consistent with the design intuition of the prior. (3) More languages and data allow for more exploration.
>
> *[Conclusion and future work]* Our method is advantageous in most cases across language numbers and data abundance. We are pioneering in the design and incorporation of linguistic priors in MMT with the SMoE model. We acknowledge that there is a lot of follow-up research that can be done in these promising directions, which we will explore in future work. For example, we can further investigate how to improve the capacity and performance of $\texttt{xx-en}$ routing exploration.
>
> **[CONS4 Writing Details]**
> Thanks for your careful reading. We have addressed the following issues in our updated draft, highlighted in orange:
> - Citation formats and grammatical consistency in the last paragraph of the "Related Work" section.
> - In Section 4.2, "Table 1" is cited as "Table 4.2".
> - The confusion of alpha and beta values is because they represent different hyperparameters. Eq. (2) is the ratio of load balancing and language grouping loss in the total loss. while in the implementation are the hyperparameters for Adam. To avoid confusion, we have changed the coefficients of the auxiliary losses to $c_1$ and $c_2$. For the Adam hyperparameters in the implementation details, $\alpha$ and $\beta$ are changed to $\beta_1$ and $\beta_2$.

---

> ### Author Response · Authors · 2023-11-21
> **Sincerely expecting further discussions from Reviewer xU8U**
>
> Dear Reviewer **xU8U**,
>
> We thank reviewer **xU8U** for the time of reviewing and the constructive comments. We really hope to have a further discussion with reviewer **xU8U** to see if our response solves the concerns.
>
> In response, we have (1) presented additional experimental results and analysis of dynamic routing; (2) provided details about model efficiency; (3) clarified other analysis and writing confusions;
>
> We genuinely hope reviewer **xU8U** could kindly check our response. Thank you!
>
> Best wishes,
>
> Authors

---

> ### Comment · Reviewer_xU8U · 2023-11-22
> **Thanks for your responses**
>
> Thank you for your comprehensive classification and the additional information. The modifications made in the manuscript have addressed all of my concerns, and I have revised my revaluation accordingly.

---

> > ### Author Response · Authors · 2023-11-22
> > **Thank you for increasing the score**
> >
> > Dear Reviewer **xU8U**,
> >
> > We sincerely appreciate the effort Reviewer **xU8U** have dedicated to reviewing our work.
> >
> > Once again, we are truly grateful for your time and support, and for increasing our score!
> >
> > Best wishes,
> >
> > Authors

---

### Official Review · Reviewer_mayu · 2023-11-01

**Soundness:** 2 fair
**Presentation:** 4 excellent
**Contribution:** 3 good
**Rating:** 6
**Confidence:** 4

**Summary:**

This paper proposes a modification of sparse mixture-of-experts (SMoE) for multilingual MT that makes routing dependent on language-specific representations (and hierarchical), and automatically adjusts the number of activated experts for each language. These two modifications are introduced to more explicitly allow for grouping by languages based on their similarity, and for allocating experts based on complexity. The new SMoE variant is tested on a range of subsets of the OPUS-100 dataset and compared against various previous SMoE variants. A set of ablations further investigates the functionality of the expert allocation strategies.

Score was raised after author response.

**Strengths:**

- The hypothesis is convincing and the design of the methods follows the hypothesis and is thoughtfully laid out and is original.
- The set of empirical evaluations and ablations is rich (if one disregards the problems with the benchmark, see below) and sound, aligns well with the hypothesis, and presents the proposed SMoE solution as an empirically successful method and favourable in comparison to other SMoE variants.

**Weaknesses:**

- Empirical comparisons:
   - [authors clarified: reimplementation] The baseline scores for the Dense model on OPUS100 are much lower than what was reported in previous papers, in particular Zhang et al. 2021 (https://openreview.net/pdf?id=Wj4ODo0uyCF), even though it’s supposed to be the same architecture and training data and evaluation metric. For example, Zhang et al.’s M2O model scores on average 29.27 BLEU, while this paper reports 25.39 in Table 2, analogously for O2M 20.93 vs 19.03. Perhaps I missed an obvious difference in modeling that could explain the difference in results, but it is questionable whether the dense baseline was tuned sufficiently, and the same for all sparse baselines.
  - [authors promised more results on OPUS100] The prominently discussed results are reported on a subset of the languages in OPUS100 (Table 1), and the two most competitive baselines (ST-SMoE, Residual-SMoE - the two baselines with 100% win ratio) are left out for the full evaluation on all languages (Table 2). This leaves the competitiveness of the proposed solution in a realistically large multilingual setup to be questioned.
- [authors provided WMT benchmark] Choice of benchmark: Unfortunately, OPUS100 is designed in such a way that different languages contain data for different domains, both in training and in test sets. This arises from the fact that many low-resource languages are only covered by religious datasets or tech localization datasets on OPUS. Therefore, there is a relatively strong domain interference, where higher resource languages also have more complex and diverse data, and lower resource languages have less complex and more repetitive data. In past work (Kreutzer et al. 2021 (https://arxiv.org/pdf/2110.06997.pdf), this was already suspected to interfere with language-as-a-task modeling), and it also explains why low-resource languages in the aggregated results often have higher avg BLEU than mid-resource languages (cf. Table 2), as they have simpler test sentences with less domain diversity. The effect seems less drastic with the selected subset of 16 languages (Table 1), but this is a less realistic setup in general.
The similarity between domains can interfere with language hierarchies that are supposed to be modeled here, i.e. data points sampled from distant languages but from the same domain can potentially be more similar to each other than data points from similar languages but different domains. This might blur the linguistics-guided routing visualization in 4.3 - it could be that routing was rather based on domains than languages, or a combination of both. With this dataset, it is unfortunately impossible to tell.
Furthermore, the amount of data per language is therefore not a suitable (sufficient) metric for task complexity or linguistic difficulty as argued in 4.3.
I would strongly recommend the authors to redo the experiments on a domain-controlled multilingual benchmark, such as a combination of WMT datasets across languages, as e.g. in Cheng et al. 2022 (https://arxiv.org/pdf/2203.07627.pdf). It might be a lot of effort, but eventually worth it to present the success of the proposed method with less interference.

**Questions:**

- [answered] Are the baseline models reimplemented? Are they coded in the same codebase as the proposed model?
- [answered] Can you quantify the similarity of routing patterns within language groups with cosine similarity in 4.3 as you did in Figure 3? Visual inspection is still hard.

---

> ### Author Response · Authors · 2023-11-20
> **Responses to Reviewer mayu [Cons1-2]**
>
> Many thanks to reviewer **mayu** for acknowledging that our designs and experiments are "convincing" and "thorough". We sincerely appreciate all constructive suggestions that help us to further improve our work. To address reviewer **mayu**'s questions, we provide point-by-point responses below.
>
> **[CONS1 Baseline Score]**
> We respectfully argue that our setting is fair and reasonable from a training cost, training supervision, and implementation detail perspective, with further analysis of possible reasons for the difference in baseline score.
>
> *[High computation cost and model convergence observation]* Our training setting is carefully designed by calculating the training cost and experiment to examine model convergence. For example, it costs about $1093.93$ A600 GPU hours to train a vanilla transformer-based SMoE model with 32 experts on the full OPUS-100 dataset (107M sentence pairs) for 500k steps (as in the CLSR [1]) with 8192 max tokens per batch (in our setting). Meanwhile, we observe that the models converge much faster than at the end of training. First, we perform a preliminary experiment training $\textit{GS-SMoE}$ and $\textit{Dense}$ in the same setting as above for 10 epochs (about 918k steps). We find that both the dense and SMoE baselines converge quickly at 100K steps, and then the loss decreases at a much slower rate. Thus, we ultimately design the 200K training steps for OPUS-100 datasets to reduce the training cost while obtaining a relatively well-trained model. A similar observation applies to OPUS-16/50 as well.
>
> *[Implementation details and comparison]* To ensure a fair comparison, we reimplement all baselines in our code base, which is based on the MoE branch of fairseq (https://github.com/facebookresearch/fairseq/tree/moe). All models are trained and evaluated using the same settings. This also answers the reviewer's question about our implementation details.
>
> *[Other possible reason]* Aside from training time, there are a few possible reasons for the difference in BLEU score, such as training framework, tokenization, and evaluation metric. For example, CLSR [1] uses tensorflow as the training framework. Some other works use different tokenizers and BLEU score implementation [2]. In particular, we adopt the same evaluation settings as in SCoMoE [3].
>
> [1] Share or Not? Learning to Schedule Language-Specific Capacity for Multilingual Translation
>
> [2] No Language Left Behind: Scaling Human-Centered Machine Translation
>
> [3] SCoMoE: Efficient Mixtures of Experts with Structured Communication
>
> **[CONS2 More Baseline Experiments in OPUS-50/100 Settings]**
> Thanks for the question. This could be explained by the results of OPUS-16 and our experimental design. We will also implement $\textit{ST-SMoE}$ in OPUS-50/100 and include it in the final version of the paper.
>
> *[Experiment design]* First, we consider a powerful baseline in the experiments with more languages. From Table 1, the higher performance of $\textit{Residual-SMoE}$ over dense and other SMoE baselines proves the effectiveness of adding a shared expert for the token-routing SMoE model. Although our $\textit{LGR-SMoE}$ performs better, we consider it as a backbone to incorporate and testify the residual technique on language-guided routing mechanism, which is named $\textit{LGR}^{res}\textit{-SMoE}$. Then, we experiment with the two language-guided routing baselines ($\textit{LGR-SMoE}$, $\textit{LGR}^{res}\textit{-SMoE}$) as shown in Table 2.
>
> *[SMoE baseline selection]* From Table 1, the average BLEU score of $\textit{ST-SMoE}$ falls behind $\textit{Residual-SMoE}$, so we focus on examining the most competitive baseline technique, Residual expert, in more language settings.
>
> *[Further experiments.]* To address the reviewer's concerns, we will train and evaluate $\textit{ST-SMoE}$ on more OPUS-50/100. However, running the full data training costs about $436.25$ A6000 GPU hours. Due to the limited time in the rebuttal period, we will include it in the final version.

---

> ### Author Response · Authors · 2023-11-20
> **Responses to Reviewer mayu [Cons3-4]**
>
> **[CONS3 Benchmark Selection]**
>
> *[Extra experiment with domain consistent dataset]* As the reviewer acknowledges, it is challenging to train models on WMT, since the data size of its usual setting [1] is about 400M combining en-xx and xx-en directions, which is four times larger than OPUS-100. So, we re-examine language guided routing $\textit{LGR-SMoE}$ on a subset of WMT to exclude possible interference from data domain similarity. Specifically, we collect all WMT16 data, lv-en from WMT17, ro-en from WMT18, and kk-en from WMT19, in total 9 language pairs. We implement data pre-processing and model training with the same setting as OPUS-16. The average BLEU scores on each language pair are presented in Table S1, which shows that language-guided routing consistently outperforms.
>
> Table S1 Performance comparisons of $\textit{LGR-SMoE}$ compared to $\textit{Dense}$ and vanilla SMoE $\textit{GS-SMoE}$ on WMT dataset. Average BLEU scores for all language pairs are reported.
> |                                 | All       | cs-en     | de-en     | et-en | fi-en | kk-en    | lv-en     | ro-en     | ru-en     | tr-en     |
> | ------------------------------- | --------- | --------- | --------- | --------- | --------- | -------- | --------- | --------- | --------- | --------- |
> | $\textit{Dense}$                | 15.62     | 18.22     | 27.06     | 15.67     | 14.04     | 3.90     | 9.98      | 22.81     | 17.73     | 11.18     |
> | $\textit{GS-SMoE}$              | 17.69     | 20.40     | 29.59     | 17.47     | 15.81     | 5.74     | 11.30     | **25.28** | **20.38** | 13.20     |
> | $\textit{LGR-SMoE}$ | **18.05** | **20.96** | **30.09** | **17.51** | **16.03** | **6.75** | **11.83** | 25.10     | **20.38** | **13.77** |
>
> *[Extra experiment with complex language difficulty metric]* We address the confusion by (1) experimenting with dynamic routing including both grammar complexity and data abundance, and (2) explaining the reason we do not include grammar complexity in the main content.
> - we have developed a variant of dynamic routing that accounts for both grammar complexity and data volume. As shown in Table S2, it consistently outperforms the dense and SMoE baselines.
> - Specifically, we group languages according to the same ratio of high/medium/low resource language numbers, with the language difficulty metrics that add grammar complexity scores (rated by GPT-4 from 1-5) and data abundance scores (0,1,2 for high/medium/low resource). Then we train $\texttt{Lingual-SMoE}^{+grammar}$ with language difficulty groups and compare it to the dense and SMoE baselines. All models are trained in OPUS-16 settings.
> - Due to the subjective characteristic of grammar complexity, we choose to use only data abundance as a metric of language difficulty. To avoid misunderstanding, we revise the paper draft by explicitly defining language difficulty in the abstract and introduction, and detailing the implementation of the method. We also update the probe test in the Appendix.
>
> Table S2 Performance comparisons of $\texttt{Lingual-SMoE}$ with dynamic routing according to language difficulty defined by grammar complexity and data abundance, compared to $\textit{Dense}$ and vanilla SMoE $\textit{GS-SMoE}$. Average BLEU scores for each direction are reported.
> | | all | en-xx | high | mid | low | xx-en | high | mid | low |
> | ------------------------------- | --------- | --------- | --------- | --------- | --------- | --------- | --------- | --------- | --------- |
> | $\textit{Dense}$ | 28.79 | 26.92 | 25.37 | 39.12 | 14.78 | 30.67 | 28.81 | 39.24 | 24.21 |
> | $\textit{GS-SMoE}$ | 30.29 | 28.28 | 25.55 | 40.71 | 18.97 | 32.31 | 28.77 | 41.70 | 29.25 |
> | $\texttt{Lingual-SMoE}^{+grammar}$ | **32.55** | **31.36** | **27.05** | **45.82** | **23.59** | **33.73** | **29.95** | **43.61** | **30.66** |
>
> **[CONS4 visualization]** For better visualization, we provide the heatmap of cosine similarity of routing decisions in the final encoder and decoder SMoE layer, where the observation is consistent with the LGR design: closer languages have higher similarity in routing. We also enlarge the original visualizations and move them to Appendix. Please refer to Figure A6-8 in the updated draft.

---

> > ### Comment · Reviewer_mayu · 2023-11-22
> > **Response**
> >
> > Apologies for the late reply. I appreciate your detailed clarifications, the updates in the draft and the additional results on WMT.
> > I'm happy to see LGR-SMoE also perform well on WMT. Significance tests would furthermore be helpful to understand the difference in scores between GS-SMoE and LGR-SMoE. With regards to previous results, it would be important to be transparent on the re-implementation and the sacrebleu configurations, so that this paper hopefully contributes to a better comparability of these methods moving forward.
> > With an optimistic spirit I'll increase my score - hoping that WMT and OPUS-100 results will be not only a side experiment in the final paper, but at the core of the reasoning of the success of the model (even if results are not as great/clear as for OPUS-16/50).

---

> > > ### Author Response · Authors · 2023-11-22
> > > **Thank you for increasing the score**
> > >
> > > Dear Reviewer **mayu**,
> > >
> > > Many thanks for the helpful feedback and positive reevaluation. We sincerely appreciate reviewer **mayu** for increasing our score.
> > >
> > > In our final version, we will further refine and improve our paper based on all reviewer comments, to include more experiments and analyses. For the reproductivity, the sacrebleu configuration is `BLEU Signature: nrefs:1 | case:mixed | eff:no | tok:13a | smooth:exp | version:2.3.1`, which has been added in the updated draft. The core implementation codes are provided in the supplement with training and evaluation scripts, and we plan to release an additional code repository in the final version.
> > >
> > > We are again very thankful for your time and support!
> > >
> > > Best wishes,
> > >
> > > Authors

---

> ### Author Response · Authors · 2023-11-21
> **Sincerely expecting further discussions from Reviewer mayu**
>
> Dear Reviewer **mayu**,
>
> We thank reviewer **mayu** for the time of reviewing and the constructive comments. We really hope to have a further discussion with reviewer **mayu** to see if our response solves the concerns.
>
> In response, we have (1) clarified our baseline implementations; (2) presented additional experimental results and analysis of dynamic routing; (3) improved visualization results;
>
> We genuinely hope reviewer **mayu** could kindly check our response. Thank you!
>
> Best wishes,
>
> Authors

---

> ### Author Response · Authors · 2023-11-22
> **Sincerely expecting further discussions from Reviewer mayu**
>
> Dear Reviewer **mayu**,
>
> We thank reviewer **mayu** for the time of reviewing and the constructive comments. We really hope to have a further discussion with reviewer **mayu** to see if our response solves the concerns.
>
> In response, we have (1) clarified our baseline implementations; (2) presented additional experiments about more benchmark and dynamic routing; (3) improved visualization results;
>
> We genuinely hope reviewer **mayu** could kindly check our response. Thank you!
>
> Best wishes,
>
> Authors

---

### Official Review · Reviewer_BJsk · 2023-11-08

**Soundness:** 4 excellent
**Presentation:** 4 excellent
**Contribution:** 4 excellent
**Rating:** 8
**Confidence:** 5

**Summary:**

The authors propose Lingual-SMoE, an MoE model suited to multilingual MT that overcomes limitations of previous work. It features:
1) Has hierarchical gating that leverages linguistic groupings of the languages on which the model is trained.
2) Uses a Dynamic Expert Allocation during training to determine the correct number of experts to allocate to each target language.

They conduct experiments on the OPUS-100 dataset with 16-100 languages, and extensively compare Lingual-SMoE to the baselines.

**Strengths:**

- The authors did several experiments to justify their design choices and overcome the limitations of previous work.
- I think the Dynamic Expert Allocation technique is intriguing and intuitive, and the results+analysis seem good. (See Questions section, though)
- The authors did extensive ablations on their proposed techniques, and compared to a number of baselines.
- The analysis based on empirical results is also good, and answer most of the questions a reader would have.

**Weaknesses:**

- Notation for the baselines is confusing - eg. from saying LS-SMoE or Hybrid-SMoE is not clear which baseline you are referring to - it made understanding the tables cumbersome. I suggest using slightly longer names like Switch MoE or GShard MoE or Hybrid TaskMoE etc (and preferrably organize as a list or table)
- I think you should reframe the language complexity part as language resourcedness. It's not that any language is more or less complex, it's the amount of data available and handling under/overfitting. This will also match all the analysis and description of DEA in rest of the paper.


Some papers to cite:
- https://arxiv.org/abs/2108.05036, https://arxiv.org/abs/2208.03306 - while these papers are for multi-domain setting, I believe these are related in spirit given that they leverage differences in input to allocate experts.
- https://proceedings.mlr.press/v202/chen23aq.html - related to DEA, but for a different purpose


Minor:
- In figure 5, are medium resource languages and low resource languages mixed up? If not this is counter intuitive and worth elaborating on/doing analysis.

**Questions:**

- wrt result with DEA added in Tables 1 and 2, do you have any analysis on why adding this is worse on LRLs for OPUS-16,100 compared to OPUS-50 (ie, the trend is reversed)? is, say, the number of experts per low resource language suddenly worse for 16,100? are there outliers for the 50 case? a trend reserval of +/-0.2 is not bad, but this is an order of magnitude reversal of the trend

- wrt Figure 4, is this equally an issue for dense models? (it's known that this is an issue in general, but it is particularly different than dense models?)
- What are the inference implications of your method? I think even theoretical trade-offs and discussion of this would be interesting, since the most related works discuss this extensively

---

> ### Author Response · Authors · 2023-11-20
> **Responses to Reviewer BJsk [Cons1-3]**
>
> We are very glad and appreciate that reviewer **BJsk** had a positive first impression. To address reviewer **BJsk's** questions, we provide point-by-point responses to your concerns as follows.
>
> **[CONS1 Baseline Notation]** Thank you for the helpful suggestions. Due to space limitations, we add a table comparing the difference between baseline models and our methods in Appendix Table A7. Also, we will find a better abbreviation for each baseline and unify the expression throughout the paper in the final version.
>
> **[CONS2 The Definition of Language Complexity]**.
>
> Thanks for the feedback. We revise and explicitly mention that we use data availability to indicate language difficulties in the methodology section of the modified draft. Revisions are highlighted in orange.
>
> In addition, we implement dynamic routing considering both grammar complexity and data amount, whose performance on OPUS-16 is shown in Table S1. Specifically, we group languages according to the same ratio of high/medium/low resource language numbers, with the language difficulty metrics that add grammar complexity scores (rated by GPT-4 from 1-5) and data abundance scores (0,1,2 for high/medium/low resource). Then we train $\texttt{Lingual-SMoE}^{+grammar}$ with language difficulty groups and compare it to the dense and SMoE baselines in OPUS-16 settings. As shown in Table S1, it consistently outperforms the dense and SMoE baselines. However, due to the subjective nature of grammar complexity, we choose not to include it in our main content.
>
> Table S1 Performance comparisons of $\texttt{Lingual-SMoE}$ with dynamic routing according to language difficulty defined by grammar complexity and data abundance, compared to $\textit{Dense}$ and vanilla SMoE $\textit{GS-SMoE}$. Average BLEU scores for each direction are reported.
> | | all | en-xx | high | mid | low | xx-en | high | mid | low |
> | ------------------------------- | --------- | --------- | --------- | --------- | --------- | --------- | --------- | --------- | --------- |
> | $\textit{Dense}$ | 28.79 | 26.92 | 25.37 | 39.12 | 14.78 | 30.67 | 28.81 | 39.24 | 24.21 |
> | $\textit{GS-SMoE}$ | 30.29 | 28.28 | 25.55 | 40.71 | 18.97 | 32.31 | 28.77 | 41.70 | 29.25 |
> | $\texttt{Lingual-SMoE}^{+grammar}$ | **32.55** | **31.36** | **27.05** | **45.82** | **23.59** | **33.73** | **29.95** | **43.61** | **30.66** |
>
> **[More References]** Thanks for the helpful references. We have included these papers in the revised draft. Specifically, we cite these papers in the "Routing Designs and Expert Capacity in SMoE" paragraph in the Related Work section, highlighted in orange.
>
> **[CONS3 Dynamic Capacity for Hard and Medium Languages]** Thanks for the detailed review. This is a plotting issue. We have updated Figure 5 in the modified draft and switched the medium and low in the legend.

---

> ### Author Response · Authors · 2023-11-20
> **Responses to Reviewer BJsk [Question1-3]**
>
> **[QUESTION1 Effect of Dynamic Expert Allocation (DEA) on Different Datasets]**
>
> We respectfully point out that the DEA has consistently improved in scale across OPUS-16/50/100. The reviewer's observation could be attributed to the implementation difference between DEA in OPUS-16 and OPUS-50/100, and the table layout.
>
> *[How to calculate the performance gain of DEA]* First, DEA is introduced to further improve the Language Guided Routing (LGR) design. Since we find that adding a shared expert is helpful for SMoE MMT, we implement a variant of the original $\textit{LGR-SMoE}$ model called $\textit{LGR}^{res}\textit{-SMoE}$ by replacing one of the input-specific experts with a shared expert. Then, DEA is added to the best model with LGR, which is $\textit{LGR}^{res}\textit{-SMoE}$ and $\textit{LGR-SMoE}$ in OPUS-50/100 setting. Thus, the improvement in DEA could be calculated by subtracting the BLEU score of $\textit{LGR}^{res}\textit{-SMoE}$ from $\texttt{Lingual-SMoE}$ in OPUS-16, and that of $\textit{LGR}$ from $\texttt{Lingual-SMoE}$ in OPUS-50/100.
>
> *[Specific performance gain of DEA on OPUS-16/50/100]* Thus, the gain of DEA in average BLEU is $0.1\%$ for OPUS-16, $0.09\%$ for OPUS-50, and $0.17\%$ for OPUS-100, which is $0.12\%$ on average.
>
> **[QUESTION2 Overfitting Problem on Low-resource Language for Dense Model]** Yes, the dense MMT model also suffers from overfitting on low-resource language, as shown by many papers on MMT with dense model [1,2]. In particular, [3] compares the dense and SMoE model overfitting existing in dense models, showing that this problem is more acute in SMoE model than the dense model, which is also the motivation of our DEA design.
>
> [1] Neural Machine Translation for Low-resource Languages: A Survey
>
> [2] Meta-Learning for Low-Resource Neural Machine Translation
>
> [3] Fixing MoE Over-Fitting on Low-Resource Languages in Multilingual Machine Translation
>
> **[QUESTION3 Model Efficiency]**
>
> We wonder whether the "inference implication" mentioned by the reviewer means inference efficiency, or the trade-off between speed and performance, as discussed in many studies of SMoE [4-6]. If the "inference implication" is inference efficiency, we record the inference average tokens processed per second (token/s) of $\textit{GS-SMoE}$ and $\texttt{Lingual-SMoE}$ on the same test set. The inference token/s for $\textit{GS-SMoE}$ is $469.84$, and that of $\texttt{Lingual-SMoE}$ is $456.83$. Compared to vanilla SMoE, our model achieves considerable improvement with marginal computational costs.
>
> [4] Task-level Mixture-of-Experts for Efficient Inference
>
> [5] MoEfication: Transformer Feed-forward Layers are Mixtures of Experts
>
> [6] Sparse MoE as the New Dropout: Scaling Dense and Self-Slimmable Transformers

---

### Author Response · Authors · 2023-11-20
**General Responses**

We greatly appreciate the time and effort that all the reviewers have put into their reviews of our paper. We extend our thanks for the valuable comments and suggestions that helped us improve our paper. In addition to the pointwise responses below, we summarize the updates that have been made to our paper.

**[Extra Experiments]**

*[Dynamic routing with extra grammar complexity information]* To address the common concern by reviewers **xU8U**, **mayu**, and **BJsk**, on dynamic routing with data amount as the only metric of language complexity, we (1) report additional experiment results of dynamic routing considering both grammar difficulty and data abundance, which consistently outperforms dense and SMoE baselines. (2) To better support our original design, we also provide an analysis of the language difficulty metrics, where we decide not to include grammar difficulty due to its subjective nature.

*[Improving $\textit{LS-SMoE}$ baseline]* Inspired by reviewer **BUYj**, we improve $\textit{LS-SMoE}$ with fixed language routing depending on both source and target languages in the encoder, and dense pre-training to better follow the original setting. The performance of the two $\textit{LS-SMoE}$ variants is still behind our $\textit{LGR-SMoE}$, which again validates the effectiveness of language-guided routing.

*[More benchmarks]* Thanks to Reviewer **mayu** for providing helpful insights into the domain discrepancy of the datasets. We test our method on a subset of WMT, and compare it to its dense and SMoE counterparts. The results verify the effectiveness of language guided routing excluding possible domain interference.

**[Extra Evaluations]**

*[More metrics]* We evaluate our $\texttt{Lingual-SMoE}$ and compare with dense and SMoE counterparts on ChrF, COMET, ROUGE, and CIDEr as complementary metrics to BLEU as suggested by **BUYj**. The evaluation results show the robustness and generalization ability of our method to variant metrics.

*[Zero-shot translation]* We test our method on zero-shot settings with unseen translation directions (@reviewer **BUYj**). Without a specific design for many-to-many translation, $\textit{LGR-SMoE}$ trained on the English-centric dataset shows better zero-shot ability than the dense model.

**[Paper Editing]**

*[Clarification on the definition of language difficulty]* To reduce confusion about the language difficulty metric, we revise the Abstract and Introduction sections to explicitly mention that we use data availability to indicate language difficulty in the Methodology section. The revisions are highlighted in orange.

*[Model efficiency]* We measure model efficiency from model size, and computational cost, to training and inference speed, as suggested by reviewers **BUYj**, **xU8U**, and **BJsk**. We report the model efficiency metrics of our $\texttt{Lingual-SMoE}$ on top of one of the current SOTA SMoE models $\textit{GS-SMoE}$. This result shows that our design improves translation performance with only marginal additional parameters.

*[Paper writing and citations]* We fix some grammatical inconsistencies, bibliography publication information, and add some new references in the revision.

*[Improving visualization of routing decision]* New cosine similarity heatmap visualization for routing decisions is updated in the appendix.

**[Reproducibility]**

The implementation codes have been provided in the supplementary with training scripts.

Hope our pointwise responses below can clarify any confusion the reviewers have. If you have any further questions, we would be happy to address them fully.

Thanks again for the efforts of all the reviewers.

---

### Meta-Review · Area_Chair_ksSt · 2023-12-14

**Metareview:**

This paper modifies the sparse mixture-of-experts (SMoE) architecture for multilingual MT to make the routing language-specific and adjusting automatically the number of activated experts for each language. The new SMoE variant is tested on a range of subsets of the OPUS-100 dataset and compared against various previous SMoE variants. Some initial concerns raised by the reviewers have been suitably addressed by the authors. All reviewers feel positively about this paper and so do I.

**Justification For Why Not Higher Score:**

Doesn't seem innovative enough to justify being distinguished as a spotlight paper.

**Justification For Why Not Lower Score:**

Proposes a new language-dependent technique which is intuitive and seems to work well in practice.

---

### Decision · Program_Chairs · 2024-01-16

Accept (poster)